# Program Synthesis Guided Reinforcement Learning for Partially Observed Environments

**Yichen David Yang**[*]
MIT EECS & CSAIL

**Jeevana Priya Inala**
Microsoft Research

**Osbert Bastani**
University of Pennsylvania

**Yewen Pu**
Autodesk Research

**Armando Solar-Lezama**
MIT EECS & CSAIL

**Martin Rinard**
MIT EECS & CSAIL

## Abstract

A key challenge for reinforcement learning is solving long-horizon planning problems. Recent work has leveraged programs to guide reinforcement learning in these settings. However, these approaches impose a high manual burden on the user since they must provide a guiding program for every new task. Partially observed environments further complicate the programming task because the program must implement a strategy that correctly, and ideally optimally, handles every possible configuration of the hidden regions of the environment. We propose a new approach, *model predictive program synthesis (MPPS)*, that uses program synthesis to automatically generate the guiding programs. It trains a generative model to predict the unobserved portions of the world, and then synthesizes a program based on samples from this model in a way that is robust to its uncertainty. In our experiments, we show that our approach significantly outperforms non-program-guided approaches on a set of challenging benchmarks, including a 2D Minecraft-inspired environment where the agent must complete a complex sequence of subtasks to achieve its goal, and achieves a similar performance as using handcrafted programs to guide the agent. Our results demonstrate that our approach can obtain the benefits of program-guided reinforcement learning without requiring the user to provide a new guiding program for every new task.

## 1  Introduction

Reinforcement learning is a prominent technique for solving challenging planning and control problems [50, 4]. Despite significant recent progress, solving long-horizon problems remains a significant challenge due to the combinatorial explosion of possible strategies. One promising approach to addressing these issues is to leverage *programs* to guide the behavior of the agents [3, 62, 39]. The approaches in this paradigm typically involve three key elements:

- **Domain-specific language (DSL):** For a given domain, the user defines a set of *components* $c$ that correspond to intermediate subgoals that are useful for that domain (e.g., "get wood" or "build bridge"), but leaves out how exactly to achieve these subgoals.

- **Task-specific program:** For every new task in the domain, the user provides a sequence of components (i.e. a program written in the DSL) that, if followed, enable the agent to achieve its goal in the task (e.g., ["get wood"; "build bridge"; "get gem"]).

- **Low-level neural policy:** For a given domain, the reinforcement learning algorithm learns an option [63] that implements each component (i.e., achieves the subgoal specified by that component). Typically a neural policy is learned as each option.

---

[*]Correspondence to yicheny@csail.mit.edu

35th Conference on Neural Information Processing Systems (NeurIPS 2021).

Given a new task in a domain, the user provides a program in the DSL that describes a high-level strategy to solve that task. The agent then executes the program by deploying the sequence of learned options that correspond to the components in that program.

A key drawback of this approach is programming overhead: for every new task (a task consists of an instantiation of an environment and a goal), the user must analyze the environment, design a strategy to achieve the goal, and encode the strategy into a program, with a poorly written program producing a suboptimal agent. Furthermore, partially observed environments significantly complicate the programming task because the program must implement a strategy that correctly, and ideally optimally, handles every possible configuration of the hidden regions of the environment.

To address this challenge, we propose a new approach, *model predictive program synthesis (MPPS)*, that automatically synthesizes the guiding programs for program guided reinforcement learning.

MPPS works with a conditional generative model of the environment and a high level specification of the goal of the task to automatically synthesize a program that achieves the goal, with the synthesized program robust to uncertainty in the model. Because the automatically generated agent, and not the user, reasons about how to solve each new task, MPPS significantly reduces user burden. Given a goal specification $\phi$, the agent uses the following three steps to choose its actions:

- **Hallucinator:** First, inspired by world-models [29], the agent keeps track of a conditional generative model $g$ over possible realizations of the unobserved portions of the environment.

- **Synthesizer:** Next, the agent synthesizes a program $p$ that achieves $\phi$ assuming the hallucinator $g$ is accurate. Since world predictions are stochastic in nature, it samples multiple predicted worlds and computes the program that maximizes the probability of success.

- **Executor:** Finally, the agent executes the options corresponding to the components in the program $p = [c_1; ...; c_k]$ for a fixed number of steps $N$.

If $\phi$ is not satisfied after $N$ steps, then the above process is repeated. Since the hallucinator now has more information (because the agent has explored more of the environment), the agent now has a better chance of achieving its goal. Importantly, the agent is implicitly encouraged to explore since it must do so to discover whether the current program can successfully achieve the goal $\phi$.

We instantiate our approach in the context of a 2D Minecraft-inspired environment [3, 57, 62], which we call "craft," and a "box-world" environment [76]. We demonstrate that our approach significantly outperforms non-program-guided approaches, while achieving a similar performance as using handcrafted programs to guide the agent. In addition, we demonstrate that the policy we learn can be transferred to a continuous variant of the craft environment, where the agent is replaced by a MuJoCo [66] Ant. Thus, our approach can obtain the benefits of program-guided reinforcement learning without requiring the user to provide a new guiding program for every new task.[2]

**Related work.** In general, program guidance makes reinforcement learning more tractable in at least two ways: (i) it provides intermediate rewards and (ii) it reduces the size of the search space of the policy by decomposing the policy into separate components. Previous research in program guided reinforcement learning demonstrates the benefits of this approach to guide reinforcement learning in the craft environment [62]. This previous research requires the user to provide both a DSL for the domain and a program for every new task. Furthermore, their approach requires that the user includes conditional statements in the program to handle partial observability, which imposes an even greater burden on the user. In contrast, we only require the user to provide a specification encoding the goal for each new task, and automatically handle partial observability.

There has been work enabling users to write specifications in a high-level language based on temporal logic [39], with these specifications then translated into shaped rewards to guide learning. Furthermore, recent work has shown that even if the subgoal encoded by each component is omitted, the program (i.e., a sequence of symbols) can still aid learning [3]. Unlike our approach, this previous work requires the user to provide the guiding programs and does not handle partial observability.

More broadly, our work fits into the literature on combining high-level planning with reinforcement learning. In particular, there is a long literature on planning with options [63] (also known as *skills* [33]), including work on inferring options [61]. Most of these approaches focus on MDPs with discrete state and action spaces and fully observed environments. Recent work [1, 41, 40, 32, 79, 74,

---

[2]The code is available at: `https://github.com/yycdavid/program-synthesis-guided-RL`

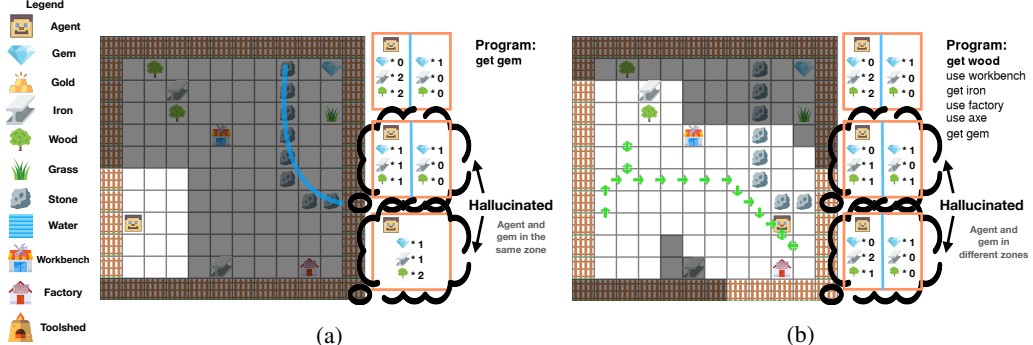

Figure 1: (a) The initial state of an example task for the craft environment. Bright regions are observed and dark ones are unobserved. This particular map has two *zones* separated by a stone boundary (blue line). The first zone contains the agent, 2 irons, and 2 woods; the second contains 1 grass and 1 gem (the goal). The agent represents the high-level structure of the map (e.g., resources in each zone) using state features. The ground truth features are in the top-right; we only show the counts of gems, irons, and woods in each zone and the zone containing the agent. The two thought bubbles below are features hallucinated by the agent based on the observed parts of the map. In both, the zone that the agent is in contains a gem, so the synthesized program is "get gem" (b) The state after the agent took 20 steps (green arrows), failed to obtain the gem, and is now re-synthesizing the program. Having explored more of the map, it predicts that the gem is in a different zone, indicated by its two hallucinations. As a result, it synthesizes a program that includes building and using an axe to break the stone, which leads to successful completion of the task.

77, 6, 64, 49] addresses the challenge of handling continuous state and action spaces by combining high-level planning with reinforcement learning to handle low-level control, but does not handle the challenge of partial observations, whereas our work tackles both challenges.

Classical STRIPS planning [24] cannot handle uncertainty in the realization of the environment. Replanning [60] can be used to handle small changes to an initially known environment, but cannot handle environments that are initially completely unknown. There has been work on hierarchical planning in POMDPs [11, 67], but this research does not incorporate predicate abstractions (i.e., state features) that can be used, for example, to handle continuous state and action spaces. Given multiple possible environments, generalized planning [27, 36, 59, 34] can be used to compute a plan that is valid for all of them. However, in our setting, oftentimes no such plan exists. We instead synthesize a plan that is valid in a maximal number of hallucinated environments. There is also prior work on planning in partially observable environments [9, 18]. Unlike our approach, these approaches assume that the effective state space is small, which enables them to compile the problem into a concrete POMDP which can be efficiently solved using POMDP algorithms. We leverage program synthesis [58] with the world models approach [29] to address these issues; generally speaking, our solver-aided plan synthesis approach is more flexible than existing planning algorithms that target narrower problem settings.

Finally, there has broadly been recent interest in using program synthesis to learn programmatic policies that are more interpretable [71, 72, 38], verifiable [8, 70, 2], and generalizable [37]. In contrast, we are not directly synthesizing the policy, but a program to guide the policy. Appendix C discusses additional related work in a broader context.

## 2 Motivating Example

Figure 1a shows a 2D Minecraft-inspired crafting game. In this grid world, the agent can navigate and collect resources (e.g., wood), build tools (e.g., a bridge) at workshops using collected resources, and use the tools to traverse obstacles (e.g., use a bridge to cross water). The agent can only observe the $5 \times 5$ grid around its current position; since the environment is static, any previously observed cells remain visible. A single task consists of a randomly generated map (i.e., the environment) and goal (i.e., obtain a certain resource or build a certain tool). We consider the meta-learning setting [25]:

we have a set of training tasks for learning the policy, and our goal is to have a policy that works well on new tasks occurring in the future.

**DSL.** A premise of our approach is a user-provided DSL consisting of components useful for the domain. Figure 2a shows the DSL for the craft environment. For each component, the user also specifies what the component is expected to achieve as a logical predicate. To deal with high-dimensional state spaces, the logical predicates are expressed over features $\alpha(s)$ of the state—e.g., the logical predicate for "get wood" is

$$\forall i, j \; . \; (z^- = i \wedge z^+ = j) \Rightarrow (b^-_{i,j} = \text{connected}) \wedge (\rho^+_{j,\text{wood}} = \rho^-_{j,\text{wood}} - 1) \wedge (\iota^+_{\text{wood}} = \iota^-_{\text{wood}} + 1).$$

This predicate is over two sets of features: (i) features $\alpha(s^-)$, denoted by a $-$, of the initial state $s^-$ (i.e., where execution of the component starts), and (ii) features $\alpha(s^+)$, denoted by a $+$, of the final state $s^+$ (i.e., where the subgoal is achieved and execution of the component terminates). The first feature is the categorical feature $z$ that indicates the zone containing the agent. In particular, we divide the map into zones that are regions separated by obstacles such as water and stone—e.g., the map in Figure 1a has two zones: (i) the region containing the agent, and (ii) the region blocked off by stones. Now, the feature $b_{i,j}$ indicates whether zones $i$ and $j$ are connected, $\rho_{i,r}$ denotes the count of resource $r$ in zone $i$, and $\iota_r$ denotes the count of resource $r$ in the agent's inventory.

Thus, this formula says that (i) the agent goes from zone $i$ to $j$, (ii) $i$ and $j$ are connected, (iii) the count of wood in the agent's inventory increases by one, and (iv) the count of wood in zone $j$ decreases by one. Appendix A.1 describes the full set of components we use.

**Approach.** Before solving any new tasks, for each component $c$, we use reinforcement learning to train an option $\tilde{c}$ that attempts to achieve the subgoal encoded by $c$. Given a new task, the user specifies the goal of the task as a logical predicate $\phi$. Encoding the goal is typically simple; for example, the goal of the task in Figure 1a is getting gem, which is encoded as $\phi := \iota_{\text{gem}} \geq 1$. Then the agent attempts to solve the task as follows.

First, based on the observations so far, the agent uses a hallucinator $g$ to predict multiple potential worlds, each of which represents a possible realization of the full map. Rather than predicting concrete states, it suffices to predict the state features. For instance, Figure 1a shows two samples of the world predicted by $g$; here, the only values it predicts are the number of zones in the map, the type of the boundary between the zones, and the counts of the resources and workshops in each zone. In this example, the first predicted world contains two zones, and the second contains one zone. Note that in both predicted worlds, there is a gem located in same zone as the agent.

Next, the agent synthesizes a program $p$ that achieves the goal $\phi$ in a maximal number of predicted worlds. The synthesized program in Figure 1a is a single component "get gem," which refers to searching the current zone (or zones already connected with the current zone) for a gem. Note that this program achieves the goal for the predicted worlds shown in Figure 1a.

Finally, the agent executes the program $p = [c_1; ...; c_k]$ for a fixed number $N$ of steps. In particular, it executes the policy $\pi_\tau$ of option $\tilde{c}_\tau = (\pi_\tau, \beta_\tau)$ corresponding to $c_\tau$ until the termination condition $\beta_\tau$ holds, upon which it switches to executing $\pi_{\tau+1}$. In our example, there is only one component "get gem," so it executes the policy for this component until the agent finds a gem.

In this case, the agent fails to achieve its goal $\phi$ since there is no gem in its current zone. Thus, it repeats the above process. Since it now has more observations, $g$ more accurately predicts the world—e.g., Figure 1b shows the intermediate step when the agent re-plans. Note that it now correctly predicts that the only gem is in the second zone. Thus, the newly synthesized program is

$$p = [\underbrace{\text{get wood}; \text{use workbench}; \text{get iron}; \text{use factory};}_{\text{for building axe}} \text{use axe}; \text{get gem}].$$

That is, it builds an axe to break the stone so it can get to the zone containing the gem. Finally, the agent executes this new program, which successfully finds the gem.

## 3 Problem Formulation

**POMDP.** We consider a partially observed Markov decision process (POMDP) with states $\mathcal{S} \subseteq \mathbb{R}^n$, actions $\mathcal{A} \subseteq \mathbb{R}^m$, observations $\mathcal{O} \subseteq \mathbb{R}^q$, initial state distribution $\mathcal{P}_0$, observation function $h : \mathcal{S} \to \mathcal{O}$,

$$
\begin{array}{rcl}
C & := & \text{get } R \mid \text{use } T \mid \text{use } W \\
R & := & \text{wood} \mid \text{iron} \mid \text{grass} \mid \text{gold} \mid \text{gem} \\
T & := & \text{bridge} \mid \text{axe} \mid \text{ladder} \\
W & := & \text{factory} \mid \text{workbench} \mid \text{toolshed}
\end{array}
$$

(a)

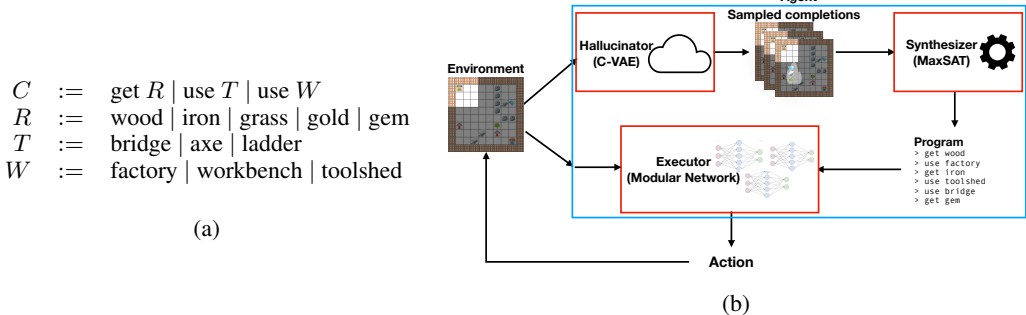

(b)

Figure 2: (a) DSL of components for the craft environment; the three kinds of components are get resource ($R$), use tool ($T$), and use workshop ($W$). (b) Architecture of our agent (the blue box).

and transition function $f : \mathcal{S} \times \mathcal{A} \to \mathcal{S}$. Given initial state $s_0 \sim \mathcal{P}_0$, policy $\pi : \mathcal{O} \to \mathcal{A}$, and time horizon $T \in \mathbb{N}$, the generated trajectory is $(s_0, a_0, s_1, a_1, \ldots, s_T, a_T)$, where $o_t = h(s_t)$, $a_t = \pi(o_t)$, and $s_{t+1} = f(s_t, a_t)$. We assume the state includes the unobserved parts of the environment—e.g., in the craft environment, it represents both the entire map and the agent's current position and inventory.

We consider a meta-learning setting, where we have a set of sampled training tasks (world configurations and goal) and a set of test tasks. Our goal is to learn a policy using the training set that achieves good performance on the test set.

**User-provided components.** We consider programs $p = [c_1; \ldots; c_k]$ composed of *components* $c_\tau \in C$. We assume the user provides the set of components $C$ that are useful for the domain. Importantly, these components only need to be provided once for a domain; they are shared across all tasks in this domain. Each component is specified as a logical predicate that encodes the intended behavior of that component. More precisely, $c$ is a logical predicate over $s^-$ and $s^+$, where $s^-$ denotes the initial state before executing $c$ and $s^+$ denotes the final state after executing $c$. For instance, the component

$$
c \equiv (s^- = s_0 \Rightarrow s^+ = s_1) \wedge (s^- = s_2 \Rightarrow s^+ = s_3)
$$

says that if the POMDP is currently in state $s_0$, then $c$ should transition it to $s_1$, and if it is currently in state $s_2$, then $c$ should transition it to $s_3$. Rather than defining $c$ over the concrete states, we can define it over features $\alpha(s^-)$ and $\alpha(s^+)$ of the states in order to handle high-dimensional state spaces.

**User-provided goal specification.** The goal of each task is specified with a logical predicate $\phi$ over the final state; as with components, $\phi$ may be specified over features $\alpha(s)$ instead of concrete states. Our objective is to design an agent that can achieve any given specification $\phi$ (i.e., act in the POMDP to reach a state that satisfies $\phi$) as quickly as possible.

## 4 Model Predictive Program Synthesis

We describe here the architecture of our agent, depicted in Figure 2b. It is composed of three parts: the *hallucinator* $g$, which predicts possible worlds; the *synthesizer*, which generates a program $p$ that maximizes the probability of success according to worlds sampled from $g$; and the *executor*, which follows $p$ to act in the POMDP. These parts are run once every $N$ steps to generate a program $p$ to execute for the subsequent $N$ steps, until the user-provided specification $\phi$ is achieved.

**Hallucinator.** First, the hallucinator is a conditional generative model trained to predict the unobserved parts of the environment given the observations. To be precise, the hallucinator $g$ encodes a distribution $g(s \mid o)$, which is trained to approximate the actual distribution $P(s \mid o)$. Then, at each iteration (i.e., once every $N$ steps), our agent samples $m$ worlds $\hat{s}_1, \ldots, \hat{s}_m \sim g(\cdot \mid o)$. Our technique can work with any type of conditional generative model as the hallucinator; in our experiments, we use a conditional variational auto-encoder (CVAE) [56].

When using state features, we can have $g$ directly predict the features; this approach works since the synthesizer only needs to know the values of the features to generate a program (see below).

**Synthesizer.** The synthesizer computes a program that maximizes the probability of satisfying $\phi$:

$$p^* = \arg\max_p \mathbb{E}_{P(s|o)} \mathbb{1}[p \text{ solves } \phi \text{ for } s] \approx \arg\max_p \frac{1}{m} \sum_{j=1}^{m} \mathbb{1}[p \text{ solves } \phi \text{ for } \hat{s}_j], \quad (1)$$

where the $\hat{s}_j$ are samples from $g$. The objective (1) can be expressed as a MaxSAT problem [48]. In particular, suppose for now that we are searching over programs $p = [c_1; ...; c_k]$ of fixed length $k$. Then, consider the constrained optimization problem

$$\arg\max_{\xi_1,...,\xi_k} \frac{1}{m} \sum_{j=1}^{m} \exists s_1^-, s_1^+, ..., s_k^-, s_k^+ \cdot \psi_j, \quad (2)$$

where $\xi_\tau$ and $s_\tau^\delta$ (for $\tau \in \{1, ..., k\}$ and $\delta \in \{-, +\}$) are the optimization variables. Here, $\xi_1, ..., \xi_k$ encodes the program $p = [c_1; ...; c_k]$, and $\psi_j$ encodes the constraints that $p$ solves $\phi$ for world $\hat{s}_j$—i.e.,

$$\psi_j \equiv \psi_{j,\text{start}} \wedge \left[ \bigwedge_{\tau=1}^{k} \psi_{j,\tau} \right] \wedge \left[ \bigwedge_{\tau=1}^{k-1} \psi'_{j,\tau} \right] \wedge \psi_{j,\text{end}},$$

where (i) $\psi_{j,\text{start}} \equiv (s_1^- = \hat{s}_j)$ encodes that the initial state is $\hat{s}_j$, (ii) $\psi_{j,\tau} \equiv \left( (\xi_\tau = c) \Rightarrow c(s_\tau^-, s_\tau^+) \right)$ encodes that if the the $\tau$th component of $p$ is $c_\tau = c$, then the transition from $s_\tau^-$ to $s_\tau^+$ on step $\tau$ satisfies $c(s_\tau^-, s_\tau^+)$, (iii) $\psi'_{j,\tau} \equiv (s_\tau^+ = s_{\tau+1}^-)$ encodes that the final state of the $\tau$th step equals the initial state the $(\tau + 1)$th step, and (iv) $\psi_{j,\text{end}} \equiv \phi(s_j^+)$ encodes that the final state of the last component should satisfy the user-provided goal $\phi$. We use a MaxSAT solver to solve (2) [16]. Given a solution $\xi_1 = c_1, ..., \xi_k = c_k$, the synthesizer returns the corresponding program $p = [c_1; ...; c_k]$.

We incrementally search for longer and longer programs, starting from $k = 1$ and incrementing $k$ until either we find a program that achieves at least a minimum objective value, or we reach a maximum program length $k_{\text{max}}$, at which point we use the best program found so far.

**Executor.** For each user-provided component $c \in C$, we use reinforcement learning to learn an option $\tilde{c} = (\pi, \beta)$ that executes the component, where $\pi : \mathcal{O} \to \mathcal{A}$ is a policy and $\beta : \mathcal{O} \to \{0, 1\}$ is a termination condition. The executor runs the synthesized program $p = [c_1; ...; c_k]$ by deploying each corresponding option $\tilde{c}_\tau = (\pi_\tau, \beta_\tau)$ in sequence, starting from $\tau = 1$. In particular, it uses action $a_t = \pi_\tau(o_t)$ at each time step $t$, where $o_t$ is the observation on that step, until $\beta_\tau(o_t) = 1$, at which point it increments $\tau \leftarrow \tau + 1$. It continues until either it has completed running the program ($\beta_k(o_t) = 1$), or after $N$ steps. In the former case, by construction, the goal $\phi$ has been achieved, so the agent terminates. In the latter case, the agent iteratively reruns the hallucinator and the synthesizer based on the current observation to get a new program. At this point, the hallucinator likely has additional information about the environment, so the new program has a greater chance of success.

## 5 Learning Algorithm

Next, we describe our algorithm for learning the parameters of models used by our agent. In particular, there are two parts that need to be learned: (i) the parameters of the hallucinator $g$ and (ii) the options $\tilde{c}$ based on the user-provided components $c$.

**Hallucinator.** The goal is to train the hallucinator $g(s \mid o)$ to approximate the actual distribution $P(s \mid o)$ of the state $s$ given the observation $o$. We obtain samples $(o_t, s_t)$ from the training tasks using rollouts from a random agent and train $g_\theta(s \mid o)$ using supervised learning. In our experiments, we take $g_\theta$ to be a CVAE and train it using the evidence lower bound (ELBo) on the log likelihood [46].

**Executor.** Our framework uses reinforcement learning to learn options $\tilde{c}$ that implement the user-provided components $c$; these options can be shared across multiple tasks. We use neural module networks [3] as the model for the executor policy; but in general our approach can also work with other types of models. In particular, we take $\tilde{c} = (\pi, \beta)$, where $\pi : \mathcal{O} \to \mathcal{A}$ is a neural module and $\beta : \mathcal{O} \to \{0, 1\}$ checks when to terminate execution. First, $\beta$ is constructed directly from $c$—i.e., it returns whether $c$ is satisfied based on the current observation $o$. Next, we train $\pi$ on the training tasks, which consist of randomly generated initial states $s$ and goal specifications $\phi$. Just for training, we use the ground truth program $p$ synthesized based on the fully observed environment; this approach

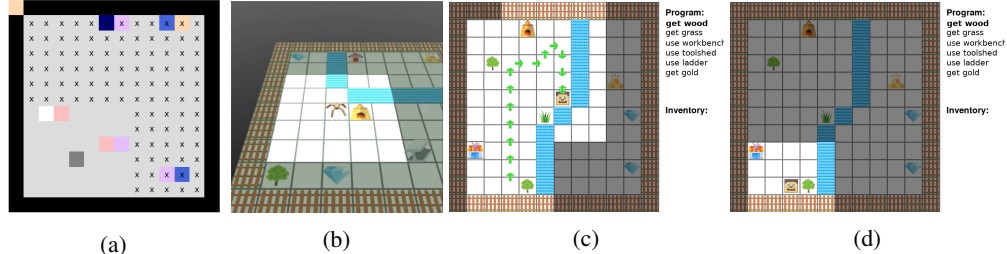

|  (a)  |  (b)  |  (c)  |  (d)  |

Figure 3: (a) The box-world environment. The grey pixel denotes the agent. The goal is to get the white key. The unobserved parts of the map is marked with "x". The key currently held by the agent is shown in the top-left corner. In this map, the number of boxes in the path to the goal is 4, and it contains 1 distractor branch. (b) The ant-craft environment. The policy needs to control the ant to perform the crafting tasks. (c,d) Comparison of behaviors between the optimistic approach (left) and our MPPS approach (right), in a task where the goal is to get gold. (c) The state when the optimistic approach first synthesizes the correct program instead of the (incorrect) one "get gold". It only does so after observing all the squares in its current zone. (d) The initial state of our MPPS strategy. It directly synthesizes the correct program, since the hallucinator knows the gold is most likely in the other zone based on the observations. Thus, the agent completes the task much more quickly.

avoids the need to run the synthesizer repeatedly during training. Given $p$, we sample a rollout $\{(o_1, a_1, r_1), ..., (o_T, a_T, r_T)\}$ by running the current options $c_\tau = (\pi_\tau, \beta_\tau)$ according to the order specified by $p$ (where $\pi_\tau$ is randomly initialized). We give the agent a reward $\tilde{r}$ at each step when it achieves the subgoal of the component $c_\tau$, as well as a final reward when it achieves the final goal $\phi$. Then, we use actor-critic reinforcement learning [47] to update $\pi$. Finally, we use curriculum learning to speed up training—i.e., we train using tasks that can be solved with shorter programs first [3].

# 6 Experiments

We empirically show that our approach significantly outperforms prior approaches that do not leverage programs, and furthermore achieves similar performance as an oracle given the ground truth program.

## 6.1 Benchmarks

**2D-craft.** We consider a 2D Minecraft-inspired game [3] (Figure 1a). A map is a $10 \times 10$ grid, where each grid cell is either empty or contains a resource (e.g., wood), obstacle (e.g., water), or workshop. Each task consists of a randomly sampled map, initial position, and goal (one of 10 possibilities, either getting a resource or building a tool), which typically require the agent to achieve several intermediate subgoals. In contrast to prior work, our agent does not initially observe the entire map; instead, they can only observe cells within two units. Since the environment is static, any previously observed cells remain visible. The actions are discrete: moving in one of the four directions, picking up a resource, using a workshop, or using a tool. The maximum episode length is $T = 100$.

**Box-world.** Next, we consider box-world [76], which requires abstract reasoning. It is a $12 \times 12$ grid world with locks and boxes (Figure 3a). The agent is given a key to get started, and its goal is to unlock a white box. Each lock locks a box in the adjacent cell containing a key. Lock and boxes are colored; the key needed to open a lock is in the box of the same color. The actions are to move in one of the four directions; the agent opens a lock and obtains the key simply by walking over it. We assume that the agent can unlock multiple locks with each key. The agent can only observe grid cells within a distance of 3 (as well as the previously observed cells). Each task consists of a randomly sampled map and initial position, where the number of boxes in the path to the goal is randomly chosen between 1 to 4, and the number of "distractor branches" (i.e., boxes that the agent can open but does not help them reach the goal) is also randomly chosen between 1 to 4.

More details about the environments are described in Appendix B.1

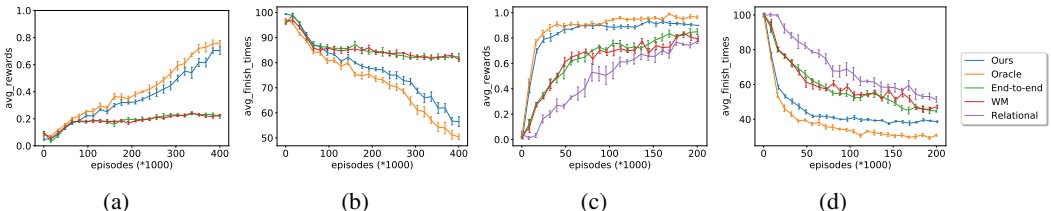

Figure 4: (a,b) Training curves for the 2D-craft environment. (c,d) Training curves for the box-world environment. (a,c) The average reward on the test set over the course of training; the agent gets a reward of 1 if it successfully finishes the task within the time horizon, and 0 otherwise. (b,d) The average number of steps taken to complete the tasks in the test set. We run all the training with 5 different random seeds, and report the mean and standard error of each metric. We show our approach ("Ours"), program guided agent ("Oracle"), end-to-end neural policy ("End-to-end"), world models ("WM"), and relational reinforcement learning ("Relational"). For our approach, we include the episodes used for training the hallucinator in the starting parts of the training curve; since the number of episodes used for hallucinator training is substantially smaller than the number of episodes for executor training, the parts for hallucinator training are hardly noticeable.

Table 1: Average rewards and average completion times on the test set for each approach at the end of training. We report the mean and standard error (in parentheses) over 5 random seeds for training.

|  | 2D-craft | | Box-world | | Ant-craft | |
|  | Reward | Finish step | Reward | Finish step | Reward | Finish step |
|---|---|---|---|---|---|---|
| End-to-end | 0.22 (0.01) | 82.3 (1.3) | 0.85 (0.02) | 44.7 (0.6) | 0.12 (0.03) | 93.1 (2.2) |
| World models [29] | 0.23 (0.01) | 81.2 (0.7) | 0.80 (0.02) | 47.2 (0.9) | 0.13 (0.01) | 91.3 (1.2) |
| Relational [76] | - | - | 0.77 (0.02) | 51.3 (1.6) | - | - |
| Ours | 0.70 (0.03) | 56.4 (2.0) | 0.90 (0.00) | 38.6 (0.4) | 0.40 (0.01) | 79.2 (1.7) |
| Oracle | 0.76 (0.02) | 50.4 (1.1) | 0.97 (0.01) | 30.8 (0.5) | 0.43 (0.02) | 77.2 (1.6) |

## 6.2 Baselines

**End-to-end.** A set of DNN policies that solves the tasks end-to-end. It uses one DNN policy per type of goal, i.e. one network will be used to solve all tasks with the goal of "get gem", another network for tasks with the goal of "build bridge". This baseline is trained using the same actor-critic algorithm and curriculum learning strategy as described in Section 5.

**World models [29].** This approach handles partial observability by using a generative model to predict the future. It trains a VAE model that encodes the current observation $o_t$ into a latent vector $z_t$, and trains a recurrent model to predict $z_{t+1}$ based on $z_1, ..., z_t$. Then, it trains a policy using the latent vectors from the VAE model and the recurrent model as inputs.

**Relational reinforcement learning [76].** For box-world, we also compare with this approach, which uses a relational module based on the multi-head attention mechanism [69] for the policy network to facilitate relational reasoning.

**Oracle.** Finally, we compare to an oracle, which is our approach but given the ground truth program (i.e., guaranteed to achieve $\phi$). This can be seen as the program-guided agent approach [62]. This baseline is an oracle since it strictly requires more information as input from the user.

## 6.3 Implementation Details

**2D-craft.** For our approach, we use a CVAE hallucinator, with MLP (with 200 hidden units) encoder/decoder, trained on 20K $(s, o)$ pairs collected by a random agent. We use the Z3 [16] solver to solve the MaxSAT problems. We use $m = 3$ hallucinated environments, $N = 20$ steps before replanning in our main experiments, and $N = 5$ in the example behaviors we show for better demonstrations. We use the same actor (resp., critic) network architecture for the policies across all

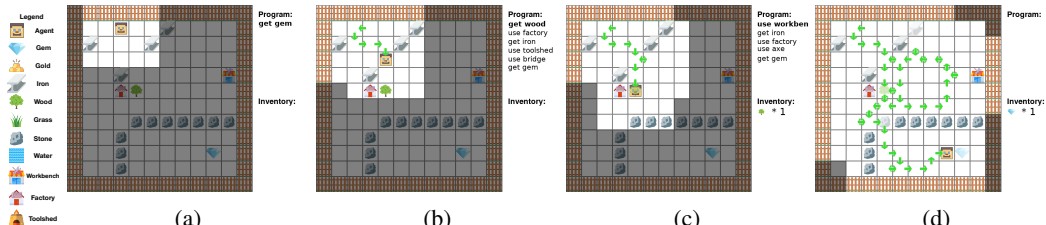

(a)          (b)          (c)          (d)

Figure 5: Example behavior of our policy in a task with the goal of getting gem. (a) The start state. The agent initially hallucinates that there is a gem in the same zone, thus starts with a simple program "get gem". (b) After several steps, the agent observes a wood and a factory. Hallucinating based on these new observations, the agent synthesizes a new program that builds a bridge to cross some water and get gem. This is a reasonable guess since wood, iron and factory are part of the recipe to build a bridge, therefore the presence of them hints that the solution might be via building a bridge. (c) After the agent finishes the "get wood" component, it observes that there are stones in the map, for which bridge cannot be used. Hallucinating based on these new observations, the agent synthesizes a new program that builds an axe to cross the stone. This is a correct program for this task. (d) The final state. The agent executes the program and successfully gets the gem.

Table 2: Comparison to optimistic synthesis and random hallucination strategies on the 2D-craft environment.

|  | Avg. reward | Avg. finish step |
|---|---|---|
| Ours | **0.70 (0.03)** | **56.4 (2.0)** |
| Optimistic | 0.42 (0.02) | 70.2 (1.2) |
| Random | 0.48 (0.02) | 72.6 (0.9) |

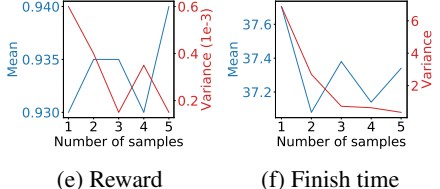

(e) Reward      (f) Finish time

Figure 6: Effect of varying the number of samples $m$ on our approach, evaluated on the box-world over 5 random seeds. Mean and variance of (a) the average reward, (b) the average finishing time on the test tasks.

approaches—i.e., an MLP with 128 (resp., 32) hidden units. We train the policies of each approach on 400K episodes over randomly sampled training tasks, and evaluate on a test set of 50 tasks. [3]

**Box-world.** Following [76], we use a one-layer CNN with 32 kernels of size $3 \times 3$ to preprocess the map across all approaches. For our approach, we have a component for each color where the subgoal is to get the key of that color; see Appendix A.2 for details. For the hallucinator, we use the same architecture as in the craft environment but with 300 hidden units, and trained with 100K $(s, o)$ pairs. For the synthesizer, we use $m = 3$ and $N = 10$. We train the policies for each approach on 200K episodes, and evaluate on a test set containing 40 tasks.

### 6.4 Results

Table 1 (left two columns) shows the performance of each approach at the end of training. Figure 4 shows the training curves. Our approach significantly outperforms the non-program-guided baselines, both in terms of fraction of tasks solved and in time taken to solve them; it also converges faster, demonstrating that program guidance makes learning significantly more tractable. Our approach also performs comparably to the oracle, delivering comparable performance with significantly less user burden. Figure 5 shows the behavior of our policy in an example task in the 2D-craft environment; see Appendix E for more examples.

**Effect of the learned hallucinator.** The hallucinator is a key in our approach to handle partial observations. Here we study the benefit of the learned hallucinator to our approach. First, we test a naive strategy for handling partial observations: the agent first randomly explores the map until the current zone is fully observed, then it synthesizes a program and follows it. This strategy only

---

[3] In our experiments, we train the hallucinator and the executor separately; but in general, one can also interleave the training of the two.

achieves an average reward of 0.024($\pm$0.004) in 2D-craft, showing that our benchmarks require effective techniques for handling partial observations. We compare to two ablations without a learned hallucinator: (i) an *optimistic* synthesizer that synthesizes the shortest possible program making best-case assumptions about the unobserved parts of the map, and (ii) a *random* hallucinator that randomly samples completions of the world (See Appendix B.3 for more details). Table 2 shows the results on the 2D-craft environment. As can be seen, our approach significantly outperforms both alternatives. Figure 3c & 3d shows the difference in behavior between our approach and the optimistic strategy; by using a learned hallucinator, our approach is able to leverage the current observations effectively and synthesize a correct program sooner.

**Effect of the number of hallucinator samples.** We vary the number of hallucinator samples $m$ on box-world. Figure 6 shows the results on the test set over 5 random seeds. As can be seen, varying $m$ does not significantly affect the mean performance, but increasing $m$ significantly reduces variance. Thus, increasing $m$ makes the policy more robust to the uncertainty in the hallucinator. This fact shows the benefit of using multiple samples and MaxSAT synthesis.

**Transfer to MuJoCo Ant.** To demonstrate that our approach can be adapted to handle continuous control tasks, we consider a variant of 2D-craft where the agent is replaced by a MuJoCo ant [53] (Figure 3b). We consider a simplified setup where we only model the movements of the ant; the ant automatically picks up resources in the grid cell it currently occupies. We focus on transfer learning from 2D-craft. In particular, we pretrain a goal-reaching policy for the ant using soft actor-critic [30]: given a random goal position, this policy moves the ant to that position. The actions output by each approach are translated into a goal position used as input to this goal-reaching policy. We initialize each policy with the corresponding model for 2D-craft and fine-tune it on ant-craft for 40K episodes. Table 1 (rightmost column) shows the results. Our approach significantly outperforms the non-program-guided baselines, both in terms of fraction of tasks solved and time taken to solve them. This demonstrates that our approach is also effective on tasks involving continuous control under a transfer learning setup.

# 7 Conclusion

We propose an approach that automatically synthesizes programs to guide reinforcement learning for complex long-horizon tasks. Our model predictive program synthesis (MPPS) approach handles partially observed environments by leveraging an approach inspired by world models, where it learns a generative model over the remainder of the world conditioned on the observations, and then synthesizes a guiding program that accounts for the uncertainty in this model. Our experiments demonstrate that MPPS significantly outperforms non-program-guided approaches, while performing comparably to an oracle given a ground truth guiding program. Our results highlight that MPPS can deliver the benefits of program-guided reinforcement learning without requiring the user to provide a guiding program for every new task.

One limitation of our approach is that, as with existing program guided approaches, the user must provide a set of components for each domain. This process only needs to be completed once for each domain since the components can be reused across tasks; nevertheless, automatically inferring these components is an important direction for future work. Finally, we do not foresee any negative societal impacts or ethical concerns for our work (outside of generic risks in improving robotics capabilities).

## Acknowledgments and Disclosure of Funding

We gratefully acknowledge support from DARPA HR001120C0015, NSF CCF-1917852, NSF CCF-1910769, and ARO W911NF-20-1-0080. The views expressed are those of the authors and do not reflect the official policy or position of the Department of Defense, the Army Research Office, or the U.S. Government. We thank the anonymous reviewers for their insightful and helpful comments.

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
