# A  Components for Environments

## A.1  Components for the Craft Environment

In this section, we describe the components (i.e., logical formulae encoding pre/post-conditions for each option) that we use for the craft environment. First, recall that the domain-specific language that encodes the set of components for the craft environment is

$$
\begin{aligned}
C &:= \text{get } R \mid \text{use } T \mid \text{use } W \\
R &:= \text{wood} \mid \text{iron} \mid \text{grass} \mid \text{gold} \mid \text{gem} \\
T &:= \text{bridge} \mid \text{axe} \mid \text{ladder} \\
W &:= \text{factory} \mid \text{workbench} \mid \text{toolshed}
\end{aligned}
$$

Also, the set of possible artifacts (objects that can be made in some workshop using resources or other artifacts) in the craft environment is

$$
A = \{ \text{ bridge, axe, plank, stick, ladder } \}.
$$

We define the following features:

- **Zone:** $z = i$ indicates the agent is in zone $i$
- **Boundary:** $b_{i,j} = b$ indicates how zones $i$ and $j$ are connected, where

$$
b \in \{\text{connected, water, stone, not adjacent}\}
$$

- **Resource:** $\rho_{i,r} = n$ indicates that there are $n$ units of resource $r$ in zone $i$
- **Workshop:** $\omega_{i,r} = b$, where $b \in \{\text{true, false}\}$, indicates whether there exists a workshop $r$ in zone $i$
- **Inventory:** $\iota_r = n$ indicates that there are $n$ objects $r$ (either a resource or an artifact) in the agent's inventory

We use $z^-, b^-, \rho^-, \omega^-, \iota^-$ and $z^+, b^+, \rho^+, \omega^+, \iota^+$ to denote the initial state and the final state for a component, respectively. Now, the logical formulae for each component are defined as follows.

**(1) "get $r$" (for any resource $r \in R$).** First, we have the following component telling the agent to obtain a specific resource $r$:

$$
\forall i, j \,.\, (z^- = i \wedge z^+ = j) \Rightarrow (b_{i,j}^- = \text{connected})
$$
$$
\wedge\, (\rho_{j,r}^+ = \rho_{j,r}^- - 1) \wedge (\iota_r^+ = \iota_r^- + 1) \wedge \mathcal{Q}.
$$

Here, $\mathcal{Q}$ refers to the conditions that the other fields of the abstract state stay the same—i.e.,

$$
(b^+ = b^-) \wedge (\omega^+ = \omega^-) \wedge (\iota_{\backslash r}^+ = \iota_{\backslash r}^-)
$$
$$
\wedge\, (\rho_{\backslash (j,r)}^+ = \rho_{\backslash (j,r)}^-),
$$

where $\iota_{\backslash r}$ means all the other fields in $\iota$ except $\iota_r$, and similarly for $\rho_{\backslash (j,r)}$. In particular $\mathcal{Q}$ addresses the *frame problem* from classical planning.

**(2) "use $r$" (for any workshop $r \in W$).** Next, we have a component telling the agent to use a workshop to create an artifact. To do so, we introduce a set of auxiliary features to denote the number of artifacts made in this component: $m_o = n$ indicates that $n$ units of artifact $o$ is made. The set of artifacts that can be made at workshop $r$ is denoted as $A_r$, and the number of units of ingredient $q$ needed to make 1 unit of artifact $o$ is denoted as $k_{o,q}$, where $q \in R \cup A$; note that $\{A_r\}$ and $\{k_{o,q}\}$ come from the rule of the game.

Then, the logical formula for "use $r$" is

$$\forall i, j \ . \ (z^- = i \wedge z^+ = j) \Rightarrow (b_{i,j}^- = \text{connected})$$

$$\wedge \ (w_{j,r} = \text{true}) \wedge \left( \sum_{o \in A_r} m_o \geq 1 \right) \wedge \left( \sum_{o \notin A_r} m_o = 0 \right)$$

$$\wedge \left( \forall q \in R, \ \iota_q^+ = \iota_q^- - \sum_{o \in A_r} k_{o,q} m_o \right)$$

$$\wedge \left( \forall q \in A, \ \iota_q^+ = \iota_q^- - \sum_{o \in A_r} k_{o,q} m_o + m_q \right)$$

$$\wedge \left( \forall o \in A_r, \ \neg \left( \bigwedge_q \iota_q^+ \geq k_{o,q} \right) \right)$$

$$\wedge \ \mathcal{Q},$$

where

$$\mathcal{Q} = (b^+ = b^-) \wedge (\omega^+ = \omega^-) \wedge (\rho^+ = \rho^-).$$

This formula reflects the game setting that when the agent uses a workshop, it will make artifacts until the ingredients in the inventory are depleted.

**(3) "use r"** ($r = $ **bridge/axe/ladder**). Next, we have the following component for telling the agent to use a tool. The formula for this component encodes the logic of zone connectivity. In particular, it is

$$\forall i, j \ . \ (z^- = i \wedge z^+ = j) \Rightarrow (b_{i,j}^- = \text{water/stone})$$

$$\wedge \ (b_{i,j}^+ = \text{connected}) \wedge (\iota_r^+ = \iota_r^- - 1)$$

$$\wedge \left( \forall i', j', \ (b_{i',j'}^+ = \text{connected}) \Rightarrow \right.$$

$$\left. \left( (b_{i',j'}^- = \text{connected}) \vee \mathcal{X} \right) \right)$$

$$\wedge \left( \forall i', j', \ (b_{i',j'}^+ \neq \text{connected}) \Rightarrow (b_{i',j'}^+ = b_{i',j'}^-) \right)$$

$$\wedge \ \mathcal{Q},$$

where

$$\mathcal{X} = (b_{i',i}^- = \text{connected} \vee b_{i',j}^- = \text{connected})$$

$$\wedge \ (b_{j',i}^- = \text{connected} \vee b_{j',j}^- = \text{connected})$$

$$\mathcal{Q} = (\omega^+ = \omega^-) \wedge (\rho^+ = \rho^-) \wedge (\iota_{\backslash r}^+ = \iota_{\backslash r}^-).$$

## A.2 Components for Box World

In this section, we describe the components for the box world. They are all of the form "get $k$", where $k \in K$ is a color in the set of possible colors in the box world. First, we define the following features:

- **Box**: $b_{k_1,k_2} = n$ indicates that there are $n$ boxes with key color $k_1$ and lock color $k_2$ in the map
- **Loose key**: $\ell_k = b$, where $b \in \{\text{true}, \text{false}\}$, indicates whether there exists a loose key of color $k$ in the map
- **Agent's key**: $\iota_k = b$, where $b \in \{\text{true}, \text{false}\}$, indicates whether the agent holds a key of color $k$

As in the craft environment, we use $b^-, \ell^-, \iota^-$ and $b^+, \ell^+, \iota^+$ to denote the initial state and the final state for a component, respectively. Since the configurations of the map in the box world can only contain at most one loose key, we add a cardinality constraint $\text{Card}(\ell) \leq 1$, where $\text{Card}(\cdot)$ counts the number of features that are true.

Then, the logical formula defining the component "get $k$" is

$$\mathcal{X} \vee \mathcal{Y},$$

where

$$\mathcal{X} = \ell_k^- \wedge \iota_k^+ \wedge (\mathrm{Card}(l^+) = 0) \wedge (b^+ = b^-)$$
$$\mathcal{Y} = (\mathrm{Card}(\iota^-) = 1) \wedge \iota_k^+ \wedge \neg\iota_k^- \wedge (l^+ = l^-) \wedge$$
$$\left(\forall k_1 . \iota_{k_1}^- \Rightarrow \left((b_{k,k_1}^+ = b_{k,k_1}^- - 1) \wedge (b_{\backslash(k,k_1)}^+ = b_{\backslash(k,k_1)}^-)\right)\right)$$

In particular, $\mathcal{X}$ encodes the desired behavior when the agent picks up a loose key $k$, and $\mathcal{Y}$ encodes the desired behavior when the agent unlocks a box to get key $k$.

# B Experimental Details

## B.1 Benchmarks

**2D-craft.** In this domain, a map is a $10 \times 10$ grid, where each grid cell is either empty or contains a resource (e.g., wood), obstacle (e.g., water), or workshop. The agent can only observe cells within the distance of 2 units. Since the environment is static, any previously observed cells remain visible. We follow the same approach as in prior work [3] to encode and preprocess the observations: each grid cell is first encoded using a one-hot encoding representing its content (with an entry for unobserved cells); then the preprocessing step extracts the $5 \times 5$ grid around the current position of the agent as the fine-scale features, and also an aggregated $5 \times 5$ grid of coarse-scale features which is aggregated over a $25 \times 25$ region from the original map (after padding) via max pooling. The flattened version of these features are the inputs to the policy networks in our approach and the baselines. More details can be found in [3] and its code repository. The test set we use contains tasks with 10 types of goals: get wood, get iron, get grass, get gold, get gem, build plank, build stick, build bridge, build axe, and build ladder. To make the test set more challenging, we include more (15 tasks) from the two hardest goals: get gold and get gem. These goals involve potentially longer horizons to achieve. The rest of the goals are in equal proportion. All our results are averaged over the test set (averaged across different types of goals). This setup follows prior work [3, 62].

For the MLP model architectures, we follow the prior work that originally introduced 2D-craft [3]; in particular, we adopt their model architecture for the actor and critic networks in both our approach and the baselines. We train our hallucinator to operate on state features (e.g. the counts of gems); it takes the state features of the observation as input and predicts the state features of the full map.

**Box-world.** In this domain, a map is a $12 \times 12$ grid with locks and boxes. The agent can only observe cells within the distance of 3 units. As in 2D-craft, since the environment is static, any previously observed cells remain visible. For encoding the observations, each grid cell is encoded using a one-hot encoding representing its content (with an entry for unobserved cells). Following [76], we use a one-layer CNN with 32 kernels of size $3 \times 3$ to preprocess the map across all approaches before feeding into the policy networks. The test set contains 40 tasks with the number of boxes in the path to the goal varying between 1 to 4; these difficulty levels are in equal proportion.

**Ant-craft.** This domain is the same as 2D-craft, except that the agent is replaced with a MuJoCo ant [53], a simulated four-legged robot. We consider a simplified setup where we only model the movements of the ant; the ant directly picks up resources, use tools, and use workshops when it is at the appropriate grid cell (e.g., we do not model the mechanics of grabbing).

## B.2 Training

We train our models on an NVIDIA GeForce GTX 1080 Ti GPU. The actor-critic training of our approach takes around a day on 2D-craft (400K episodes), 12 hours on box-world (200K episodes), and a day for fine-tuning ant-craft (40K episodes). We use the Adam optimizer [45] with a learning rate of 0.002. We use a batch size of 10 episodes.

## B.3 Ablations

Here, we provide more detail on the two ablations without a learned hallucinator.

**Optimistic synthesizer.** The optimistic synthesizer considers the unobserved parts of the world to be in any possible configuration. If a program can achieve the goal under any one of these configurations, this program is considered to be correct. The optimistic synthesizer chooses the shortest program considered to be correct in this optimistic sense. For example, if the goal of the task is "get gem", and there is some unobserved grid cells in the current zone, then an optimistic synthesizer will always synthesize the simplest program "get gem". This baseline also demonstrates the importance of using a hallucinator, instead of a heuristic such as pure optimism.

**Random hallucinator.** The random hallucinator randomly predicts the configuration of the unobserved parts of the world. In our experiments, the hallucinator directly predicts the abstract state features, so the random hallucinator simply predicts random values for each entry of the state features (e.g., number of wood in zone 1) under the condition that it does not conflict with existing observations (e.g., predicting number of wood in zone 1 to be 1 when there are already 2 woods observed in zone 1). The purpose of this ablation is to demonstrate the importance of using a learned hallucinator.

## C    Additional Related Work

**Program synthesis.** There has been a long line of work on program synthesis, which targets the problem of how to automatically synthesize a program that satisfies a given specification [55, 58, 28, 73, 31]. More broadly, recent work has explored learning neural network models to predict the program [17, 10, 14, 13, 5], as well as using neural models to guide synthesis [44, 54, 78, 7, 23, 21, 15, 51, 22]. There has also been work leveraging program synthesis to improve performance in image and natural language domains [19, 20, 68, 75, 65, 35, 12]. In contrast, our work uses program synthesis to guide reinforcement learning.

**Task and motion planning (TAMP).** TAMP is a hierarchical planning approach that uses high-level task planning and low-level motion planning [42, 26]. TAMP by itself does not handle partial observability; recent work has proposed extensions to address this challenge. For instance, [52] learns a full symbolic program to handle all possible cases—this program tends to be very complex (with many branches) and hence hard to learn. In contrast, our approach learns a simple straight line program that is most likely to solve the task and then replans if needed. Furthermore, [52] only handles discrete partial observations, whereas our approach does not have this restriction. Next, [43] performs planning in the belief space, which is more similar to our strategy. However, they make the significantly stronger assumption that a structured representation of belief space is available; in particular, they assume a probability distribution over the abstract state space is provided. In general, such a distribution can be difficult to obtain—most deep generative models are unable to explicitly provide the distribution over abstract states; instead, they provide either samples (e.g., GANs and VAEs) or probabilities of given states (e.g., normalizing flows; VAEs can provide a lower bound). As a consequence, it would be difficult to apply this approach to our environments.

## D    Additional Analysis

### D.1    Stand-alone evaluations

**Hallucinator.** We perform additional experiments that measure the prediction accuracy of our trained hallucinator for 2D-craft. We measure accuracy in two ways. The first is the percentage of cases where the predicted state features match the ground truth state features in every entry of the state feature (e.g. the number of zones is an entry, the number of wood in zone 1 is an entry). We call this the "whole" accuracy. The second is the percentage of entries that are correctly predicted, treating each entry of the state feature separately. We call this the "individual" accuracy. We measure accuracy on the test set at different number of steps into the episode. The results are shown in Table 3. As can be seen, the learned hallucinator can correctly predict many entries of the state features, but rarely predicts the whole state features perfectly. This result is due to the intrinsic randomness in the distribution $P(s \mid o)$. Note that accuracy increases with the number of steps into the episodes since the agent has explored more of the map later in the episodes.

**Executor.** We measure the success rate of the learned executor in our approach at achieving a given component. We evaluate on the test set of 2D-craft environment, focusing on components from the oracle programs. The success rate is 93.8% (so the failure rate is 6.2%). The most common failure cases are that the agent gets stuck in some local region of the map. Note that since the program for

Table 3: Standalone accuracy of the hallucinator

| Step | Whole acc. | Individual acc. |
|------|------------|-----------------|
| 0 | 0.0% | 70.9% |
| 20 | 4.5% | 82.9% |
| 40 | 4.8% | 85.5% |

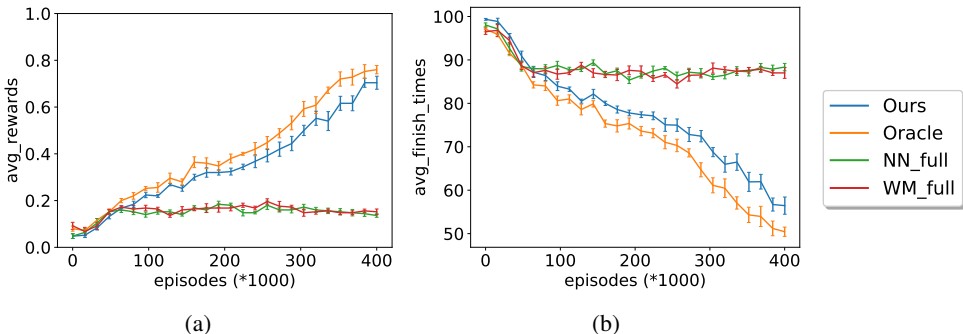

(a)                    (b)

Figure 7: Training curves for the 2D-craft environment, comparing our approach with baselines trained on fully observed environments. (a) The average reward on the test set over the course of training. (b) The average number of steps taken to complete the tasks on the test set. We run all the training with 5 different random seeds, and report the mean and standard error of each metric. We show our approach ("Ours"), program guided agent ("Oracle"), end-to-end neural policy trained on fully observed maps ("NN-full"), and world models trained on fully observed maps ("WM-full").

each task typically includes more than one component, this 6.2% failure rate will result in >6.2% failure rate in completing the tasks.

## D.2  Baselines trained with fully observed maps

In our experiments, we use the programs synthesized from the fully observed maps for training the executor in our approach. This approach avoids repeatedly running the MaxSAT synthesizer during training, which helps speed up training. To ensure this additional information is not responsible for the performance of our approach compared with the non-program-guided baselines, we perform an additional experiment that trains the baselines in fully observed environments. Figure 7 shows results for the 2D-craft environment. As can be seen, our approach continues to significantly outperform the non-program-guided baselines. These results show that providing fully observed map information during training is not the reason our approach outperforms the baselines.

## D.3  Non deterministic environment

We perform an additional experiment to study how our approach works when the environment is non-deterministic. We create a non-deterministic version of 2D-craft, where each action has 20% chance of failing (when a move action fails, the agent move to a random direction; when a use action fails, the action becomes a no-op). Table 4 shows the results. As can be seen, all the approaches take a longer time to solve tasks in these non-deterministic environments, but our approach continues to significantly outperform the non-program-guided baselines and perform comparably to the oracle. For the non-program guided baselines, the ratio of test tasks successfully solved does not change significantly, likely because they fail to solve the challenging tasks even when the environment is deterministic.

## E  Additional Examples

Table 4: Performance on the test set for the non-deterministic version of 2D-craft

|  | Avg. reward | Avg. finish step |
| --- | --- | --- |
| End-to-end | 0.22 (0.02) | 83.3 (1.8) |
| World models | 0.20 (0.01) | 83.6 (0.7) |
| Ours | 0.47 (0.03) | 73.1 (1.2) |
| Oracle | 0.50 (0.03) | 69.9 (1.6) |

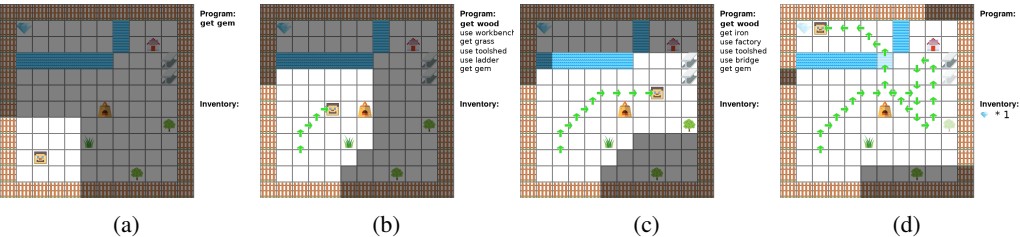

Figure 8: Example behavior of our policy in a task with the goal of getting gem. (a) The start state. The agent initially hallucinates that there is a gem in the same zone, thus starts with a simple program "get gem". (b) After several steps, the agent observes a grass and a toolshed. Hallucinating based on these new observations, the agent synthesizes a new program that builds a ladder to get gem (which requires grass and toolshed). (c) After several more steps, the agent observes some water and iron. It re-synthesizes a new program that builds a bridge to cross water. This is a correct program for this task. (d) The final state. The agent executes the program and successfully get the gem.

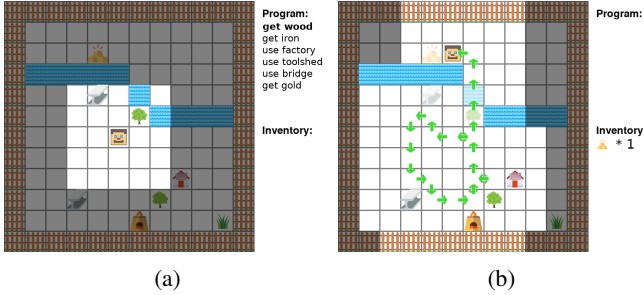

Figure 9: Example behavior of our policy in a task with the goal of getting gold. (a) The start state. By hallucinating based on the current observations, the agent correctly synthesizes a program that builds and uses a bridge to get to the other zone and get gold. (b) The final state.

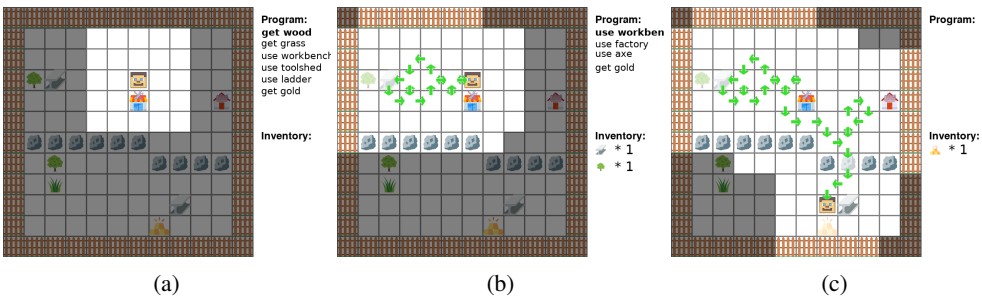

Figure 10: Example behavior of our policy in a task with the goal of getting gold. (a) The start state. Based on its hallucinations, the agent synthesizes a program that builds and uses a ladder to get a gold in the other zone. However, there is not enough resources and facilities to make a ladder in this map. (b) The intermediate state when the agent re-synthesizes a new program. With more observations, the agent changes the program to building and using an axe instead, which is a feasible solution in this map. (c) The final state.

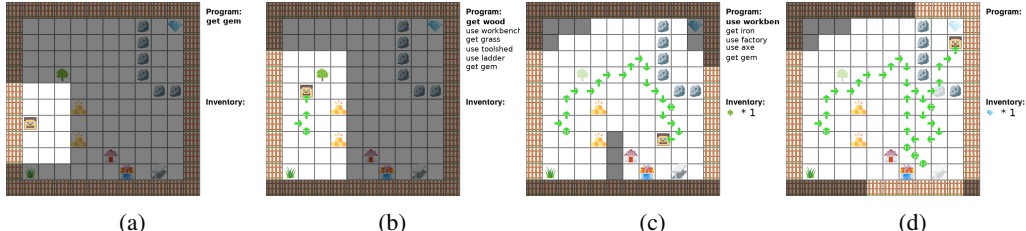

Figure 11: Example behavior of our policy in a task with the goal of getting gem. (a) The start state. The agent starts with a simple program "get gem". (b) After several steps, the agent observes a grass and a wood. Hallucinating based on these new observations, the agent synthesizes a new program that builds a ladder to get gem (which requires grass and wood). (c) During its search for workbench, the agent observes all the resources for building an axe. Therefore, it re-synthesizes a new program that builds a axe to cross the stone boundary. This is a correct program for this task. (d) The final state. The agent executes the program and successfully get the gem.