# OpenReview forum: "Program Synthesis Guided Reinforcement Learning for Partially Observed Environments"
_NeurIPS.cc/2021/Conference — NeurIPS 2021 Spotlight_

### Official Review · Reviewer_aK3Q · 2021-06-25

**Rating:** 7
**Confidence:** 5

**Summary:**

This paper aims to address the problem of solving long-horizon tasks in a partially observable environment by introducing a program-guided reinforcement learning framework. Specifically, given a task specification, this paper proposes to first synthesize a high-level instruction (i.e. a program) that consists of a sequence of subgoals and then execute the instruction to fulfill the task. To this end, the paper proposes a framework that comprises three components: (1) a generative model that can hallucinate the unobserved portions of the environment, (2) a program synthesizer that can produce a program that would fulfill the task specification, and (3) a program executor that can execute the program. The experiments show that the proposed framework outperforms the baselines including an end-to-end learning model and world models in terms of sample efficiency, obtained rewards, and the number of steps to finish a task. I believe this work studies a promising research direction (i.e. program-guided RL) and proposes an interesting framework to address the task. I would like to see this paper included in the conference while also hope the authors address my questions, including providing more comprehensive analyses are required to justify some design choices and adding missing relevant works.

**Ethical Concerns:**

I currently do not recognize any potential negative impact or ethical concerns for this work.

**Limitations And Societal Impact:**

Described in the main review section.

**Main Review:**

## Paper strengths and contributions
**Motivation**
- The motivation for solving a long-horizon task in a partially observable environment by first synthesizing a high-level plan is convincing.
- The idea of utilizing programs as a representation to describe high-level plans that need to be executed to fulfill a given task specification is intuitive. This paper presents an effective way to implement this idea.
- The limitations of prior works are explicitly mentioned, which strengthens the motivation of this work: (1) most of symbolic planning works cannot deal with continuous state and action spaces or partially observable environments, and (2) deep RL methods struggle at learning long-horizon tasks.

**Technical contributions**
- Hallucinating observable parts of the environment by predicting abstract state variables instead of concrete states seems effective.
- Formulating the problem of fulfilling a given task specification as a MaxSAT problem and utilizing a MaxSAT solver to synthesize the plan is convincing.

**Clarity**
- The overall writing is very clear. The organization is easy to follow. The authors utilize figures well to illustrate the ideas.
- Motivating Example Section not only provides a clear description of the family of problems considered in this paper but also presents the intuitions of the proposed framework. This writing flow makes the paper easier to read.

**Related work**
The authors give a clear description of the related prior works from both the perspectives of reinforcement learning and planning.

**Ablation study**
Ablation studies are helpful for understanding some design choices and the effectiveness of submodules.
- Table 6 verifies the advantage of using a learned hallucinator.
- Figure 7 justifies the proposed CVAE which allows for sampling multiple possible outcomes, while the margins are small.

**Experimental results**
- The experimental results demonstrate that the proposed framework outperforms a DRL baseline and a world model baseline and performs similarly to an oracle method (program guided agent) which utilizes a given ground truth program.
- The authors, instead of evaluating on a single domain like most of the related works, evaluated the proposed framework and the baselines in two domains (I consider the 2D-craft environment and Ant-craft environment the same), making the results more convincing.

**Reproducibility**
The code is provided, which is helpful for understanding the details of the proposed framework.

## Paper weaknesses & questions

**The performance breakdown**
Only the overall performance of the proposed framework is shown in the paper. It is not clear how well each module (i.e. the hallucinator and the executor) performs. It would be insightful to break down the overall performance.
- For the hallucinator, I would like to see how accurate (in %) it can predict the unobserved parts of the environments so that I would know better how much error comes from the prediction. With this measurement, it would be easier to justify the design choice of the hallucinator (see below).
- For the executor, I would like to see if the option policy can always complete a given component so that the failure can be attributed to either the predictions of the hallucinator or the execution.

**Design choices of the hallucinator**
Only the comparisons of the proposed hallucinator, an optimistic hallucinator, and a random hallucinator are provided in Table 6. This only justifies the necessity of learning the hallucinator. However, it is not clear if training a CVAE for this task would yield the best results. I would like to see the performance of the following:
- A deterministic baseline: instead of predicting a distribution allowing sampling, it would make sense to compare against a naive auto-encoder model that produces deterministic predictions so that we can verify the intuition of using CAVE to handle multiple possible outcomes.
- VQ-VAE: I am wondering if using a conditional VQ-VAE would yield better results especially given the state features are discrete which aligns well with VQ-VAE discrete latent codes.
I understand the tight timeline of the rebuttal so I am not demanding the abovementioned results. I simply believe including it would make the paper stronger and more convincing.

**Predict concrete states**
While it is intuitive to predict abstract feature states in this case, it would be informative to justify this choice (i.e. train a hallucinator that predicts raw states).

**Design choices of the executor**
The authors proposed to use modular networks for the executor. I wonder if it actually yields better results compared to a simple multitask policy trained to solve all options. Again, I am not asking the results for the rebuttal.

*Executor training*
Why not training the executor by simply sampling a subgoal from all the possible subgoals instead of training it to solve the entire episode, especially given that the reward signal is given whenever a subgoal is completed?

**Missing related work: program synthesis**
The related work covers two lines of works: reinforcement learning and planning. However, I feel an important line of prior works is missing: program synthesis. Program synthesis methods aim to automatically synthesize programs from high-level task specifications. The program synthesis problems considered in this work can be cast as solving MaxSAT problems with a solver. Yet, this is not always the case and this is also when program synthesis works play a role - to search a program or learn to synthesize a program. Therefore, I believe it would be important to discuss this line of work. Some example papers are as follows:
- RobustFill: Neural Program Learning under Noisy I/O
- Leveraging Grammar and Reinforcement Learning for Neural Program Synthesis
- Synthetic Datasets for Neural Program Synthesis
- Learning to Describe Scenes with Programs
- Neural Program Synthesis from Diverse Demonstration Videos
- Learning to Infer and Execute 3D Shape Programs
- Improving Neural Program Synthesis with Inferred Execution Traces
- Execution-Guided Neural Program Synthesis

**MaxSAT assumption**
This paper assumes the program synthesis problem can be cast as a MaxSAT problem, which is not always the case. I would like to know what authors think about how the proposed framework can be modified to solve problems that do not fulfill this assumption.

**Ant-craft**
Since the low-level controller of the ant is pre-trained. I am not sure what the difference between the Minecraft environment and the ant craft environment. But, maybe it is helpful for the readers who do not understand the intuition (i.e. decouple the symbolic part and the neural network policies) which allows the proposed framework to deal with continuous state and action space

**Code**
When I tried to reproduce the results, there are some missing dependencies that are not mentioned in the README, including pyyaml, matplotlib, tqdm. It would be easier to provide a requirement text file to include those.

**Minor**
Figure 6: should be Table 6

## Other metrics

### Relevance and significance
- Solid contribution to a relevant problem

### Novelty
- One idea that surprised me by its originality, solid contributions otherwise

### Technical quality
- Technically adequate for its area, solid results

### Experimental evaluation
- Sufficient evaluation w.r.t. most criteria

### Clarity
- Very clear, only minor flaws

**Time Spent Reviewing:**

11

---

> ### Author Response · Authors · 2021-08-11
> **Thank you!**
>
> First of all, we are extremely thankful for the expert reviewer to spend much of their valuable time to provide a detailed and insightful review. The reviewer provides many great suggestions on further improvements of the paper as well as interesting directions for future research. This is extremely valuable to us. Thank you!
>
> Here we present answers to reviewer's questions and the improvements we made according to the reviewer's suggestions.
>
> ### The performance breakdown
>
> **Hallucinator:**
> We have performed additional experiments that measure the prediction accuracy of our trained hallucinator for 2D-craft. We measure two types of accuracy. The first is the % of cases where the predicted state features match the ground truth state features in every entry of the state feature (e.g. the number of zones is an entry, the number of wood in zone 1 is an entry). We call this the “whole” accuracy. The second is the % of entries that are correctly predicted, treating each entry of the state feature separately. We call this the “individual” accuracy. We measure these accuracies on the test set, at different number of steps into the episode. The results are as follows:
>
> | Step | Whole Accuracy | Individual Accuracy |
> | -- | -- | -- |
> | 0 | 0.0% | 70.9% |
> | 20 | 4.5% | 82.9% |
> | 40 | 4.8% | 85.5% |
>
> We can see that the learned hallucinator can correctly predict many entries of the state features, but rarely predict the whole state features perfectly, due to the intrinsic randomness in the distribution $P(s|o)$. The accuracies increase with the number of steps into the episodes since the agent has explored more of the map later in the episodes.
>
> **Executor:**
> We measure when given a correct component (from the oracle program), the success rate of our learned executor to achieve that component. Evaluated on the test set of our 2D craft environment, the success rate is 93.8% (failure rate 6.2%). We find that the most common failure cases are because the agent gets stuck at some local region of the map. Note that since the program for each task typically includes more than one component, this 6.2% failure rate will result in >6.2% failure rate in completing the tasks. To see this, for a task including 5 components, an 80% (4 out of 5) success rate for each component will result in 0% success rate for completing the task.
>
> ### Design choices of the hallucinator
>
> We thank the reviewer for the suggestions on the alternative design choices for the hallucinator. We also think that these are interesting experiments to study. We will try our best to explore these in the future. Note that our contribution is orthogonal to the choice of the hallucinator model; our approach can work with any type of hallucinator model as long as it can learn the conditional distribution $P(s|o)$.
>
> ### Predict concrete states
>
> From some initial experiments, we found it is hard to train the CVAE to predict the concrete states well in the 2D-craft domain. We think this might be due to the raw states being high dimensional and sparse, which makes learning its distribution a harder task. We can do a more thorough comparison on this and include the results in the appendix.
>
> ### Design choices of the executor
>
> We thank the reviewer for the suggestion. We are running additional experiments on using a multitask policy for the executor (a single neural model conditioned on the option). Initial results show that it achieves a similar performance as the modular network executor. We are running more random seeds for better evaluation, and will update the results when we have them. Note that our contribution is orthogonal to the choice of the executor model; our approach can work with any type of executor model as long as it can learn an option policy for each component.
>
> **Executor training:**
> First, By training the executor policies jointly on the full episodes, we obtain training signals on how well the policy for one option affects the later parts of the episode after this option finishes (through exploration or resource collection). This enables the policies to learn to act in a way that is beneficial in the long term, in addition to achieving its own subgoal.
>
> Secondly, if we want to train each option separately, we need a way to sample the starting states for each option. When the executor is used to solve the full tasks, the options might start at different steps during the episode. Therefore, to make the executor training effective, we need the sampling distribution of the starting states for each option to match the actual distribution of the states when the option is invoked during solving full tasks. This is not straightforward to achieve, so we choose to train the options jointly over the full episodes.
>
> ### Missing related work: program synthesis
>
> We thank the reviewer for sharing this related work. We will add discussions on this line of work to our paper. These approaches largely rely on neural-guided enumerative synthesis to synthesize programs in a target domain. Enumerative synthesis is challenging in our domain due to the large number of programs; for example 2D-craft contains 1.7 million programs up to length 6, and this search space grows exponentially as program length increases. We found that MaxSAT synthesis is effective in our domains.
>
> We agree with the reviewer that in cases where the synthesis problem cannot be cast into a MaxSAT problem (and more generally a MaxSMT problem), one could use the neural-guided program synthesis approaches for synthesizing the guiding programs. Our proposed model predictive program synthesis framework can also be used in conjunction with an enumerative synthesizer (potentially neural-guided). We think this is a very interesting direction for future research. We will add a discussion to our paper.
>
> ### MaxSAT assumption
>
> The synthesis problem can be cast into a MaxSAT problem when the logical formulae for each component involves boolean variables and predicates over boolean variables. In cases where more complex state abstractions are used (e.g. involving real valued variables or intervals), instead of a MaxSAT problem, the resulting synthesis problem can be cast into a MaxSMT problem. In such cases, one can either use existing decision procedures for the theories involved in the MaxSMT problem, or write custom decision procedures in case there are no existing ones available.
>
> Another potential approach is to leverage neural program synthesis techniques to synthesize the guiding programs, as is also pointed out by the reviewer. Our proposed model predictive program synthesis framework can also be used in this case; one would use neural-guided enumerative search to search for a program that satisfies a (approximately) maximum number of hallucinated worlds. Here, one would train a neural model to predict/score guiding programs given the hallucinated states as inputs. We think this is an exciting direction for future work and thank the reviewer for these insightful comments.
>
> ### Ant-craft
>
> The reviewer’s understanding is exactly correct. The main purpose of the Ant-craft environment is to demonstrate how our approach can be used to handle continuous state and action spaces.
>
> ### Code
>
> We thank the reviewer for the suggestions. We will include the dependencies in the code repository. We deeply appreciate the reviewer for their careful review.
>
> ### Minor
>
> We thank the reviewer for the suggestions and will edit accordingly.
>
>
> We are very thankful for the insightful suggestions from the reviewer that helped us improve our paper, as well as the interesting directions for future work. We hope that our response has addressed the reviewer's questions. We are happy to discuss more if the reviewer has further questions and thoughts!

---

> > ### Comment · Reviewer_aK3Q · 2021-08-14
> > **Re: Thank you!**
> >
> > Thanks for the response, which answers my questions and addresses my concerns. I believe including the performance breakdown and the results of predicting concrete states (or explaining how it does not work) as well as incorporating other comments would make this paper stronger. At this point, I have no further questions.

---

### Official Review · Reviewer_Wt3P · 2021-07-13

**Rating:** 4
**Confidence:** 3

**Summary:**

The algorithm is set in a problem where the grid for exploration is not fully observable. The authors introduce a new approach called model predictive program synthesis (MPPS) that involves a hallucinator, a synthesizer and an executor. The hallucinator predicts the state features with CVAE, the synthesizer provides programs through optimization of a MAXSAT problem. And the executor executes the programs.


**Limitations And Societal Impact:**

I don’t think there are any ethical concerns regarding this paper. And some limitations are also addressed. For specific problems, please refer to the last section.

**Main Review:**

The paper is shown to outperforms the other comparison methods. However, I think it’s missing some discussions and comparisons that help put the problem and method in the bigger context.

I’d like to see a bit more discussion on why this particular model is able to outperforms the other methods. Is it because of the SAT solver? or the hallucinator? Or because it didn’t specify goals? A more in-depth discussion would add more weight and clarity to the paper.

How a hallucinator is trained is not very clear to me. A diagram or an example input/output would be nice. This would facilitate understanding of the method.

Explanation of the end-to-end method is not clear. Does it also have a hallucinator?

It seems like some of the maze or corridor environments in deep rl also have the same problem of not having full observability. There are already solutions exist for those types of problems without program guidance. Can you explain why program synthesis is necessary for this type of problems? Or at least are there any quantitative comparisons that show improvements or such?


**Time Spent Reviewing:**

3.5

---

> ### Author Response · Authors · 2021-08-11
> **Rationale behind good performance; Hallucinator training; End-to-end baseline; Benefit of program guided RL**
>
> We thank the reviewer for their review, and we answer the reviewer's questions here. We hope that this could help the reviewer better understand our paper.
>
> ### Rationale behind the better performance
>
> Our approach outperforms non-program guided baselines (end-to-end, world models, relational reinforcement learning). This is mainly because we utilize programs to guide the policy; the programs decompose the policy into sub-policies and specify a high-level structure of the policy. This makes reinforcement learning more efficient, and also improves the performance and generalization during inference time. The same findings are reported in prior works as well (program guided policies outperforms non-program guided policies) [1, 2].
>
> A main contribution of our paper over prior work on program guided approaches is that we remove the user burden of manually writing the guiding programs for every new task, while maintaining the benefit of program guided policies. We achieve this by our MPPS technique, which automatically synthesizes the guiding programs under partial observations.
> We will try our best to provide more detailed discussions on this topic in the paper.
>
> ### Hallucinator training
>
> First, we obtain samples of observation and ground truth state pairs $(o_t, s_t)$ using rollouts collected by a random agent on sampled training tasks. The observations are the partially observed map up to the current step; some examples of such partially observed maps can be seen in Figure 4. Since we train the hallucinator to directly predict state features, we process the state $s_t$ and observation $o_t$ to extract the abstract state features. The abstract state features include the number of zones, the count of each resource in each zone etc.. This is represented as a bit vector. The CVAE hallucinator is then trained to predict the abstract state of the full map conditioned on the abstract state of the partially observed map. This training is done in a supervised manner, using the standard evidence lower bound (ELBo) on the log likelihood (see [3] for more details).
>
> ### End-to-end baseline
>
> End-to-end does not have a hallucinator. It is a neural network policy that directly maps POMDP observations to actions.
>
> ### Why program guidance is needed for partial observability
>
> Program guidance provides significant performance gains compared to non-program guided approaches (including deep RL) for long horizon problems with sparse rewards. Prior work [1,2] has shown this in different environments. However, prior work needs the users to provide the programs manually for every new task. We focus on the problem of automatically synthesizing the guiding programs under the partial observed settings (which is more interesting and challenging than fully observed settings). Therefore, our technique can reduce the user burden while maintaining the benefit of program guidance.
>
>
> We hope that the above answers have addressed the reviewer's questions and helped the reviewer gain a better understanding. We are happy to discuss more if the reviewer has further questions.
>
> [1] Program Guided Agent, Shao-Hua Sun et.al., ICLR 2020
>
> [2] Modular multitask reinforcement learning with policy sketches, Jacob Andreas et.al., ICML 2017
>
> [3] Auto-Encoding Variational Bayes, Kingma and Welling, ICLR 2013

---

### Official Review · Reviewer_1PbL · 2021-07-16

**Rating:** 4
**Confidence:** 3

**Summary:**

The authors introduce MPPS, which stands for model predictive program synthesis, a method to leverage programs to guide reinforcement learning in partially observable problems that require planning over long-horizon. MPPS  trains a generative model to predict the unobserved portions of the world, and then synthesizes a program based on samples from this model in a way that is robust to its uncertainty. The authors illustrate the performance of their algorithm on a variety of benchmarks including a 2D Minecraft-inspired environment.

**Limitations And Societal Impact:**

This method does not add limitations or potential negative societal impact to existing reinforcement learning methods.

**Main Review:**

I warmly thank the authors for their work.

I think that the domain covered by this paper is very exciting and that this type of approach is a very promising direction for Reinforcement Learning. However, in this current form this paper exhibits several limitations:

- This paper was very difficult for me to follow. I think it would require more care to introduce the concepts, position the method with respect to the literature, and better detail the examples.

- I did not get in the introduction what is precisely the problem that the authors aim to solve and why the current methods fail to do so.

- At the very beginning of the paper, both concepts Domain-specific language and Task-specific program remain unclear to me. Especially, the example “gather wood” is used for both.

- The authors often mention that their method alleviates the user burden to provide a guiding program for every new task. What the authors mean here remains unclear to me and I believe an example would help to better understand.

- Many concepts are mentioned without being introduced first. For instance, in the very beginning, the authors talk about options without explaining what options are.

- Many mathematical notations are also not properly introduced.  For instance, on page 2 in the definition of the executor, p = c 1 ; ...; c k, the variables c_1, …, c_k are not introduced and I do not understand what they refer to.

- Section 2 covers a motivating example that remains unclear to me. I think that this section is very abstract for a motivating example and that most of the mathematical notations might be kept for later in the paper and replaced here by concrete examples.

- In the related work, the authors missed the literature about Neural Programmer Interpreters and notably AlphaNPI that is also an RL method guided by programs that would be relevant as a baseline in this study.

- The authors mention several times options but do not cover them properly in the related nor compare to them. I think that at least the Option-Critic and HAC should be mentioned, and even compared to, in this study. I understand the point made by the authors that work over options does not cover the partially observed setting however, a simple baseline to validate that this method is more relevant than using options might be to consider an option algorithm such as HAC and add a recurrent core to the high-level policy to handle partial observability.

- The different neural networks architecture as well as the RL agent used to train MPPS remain unclear to me. I think that the paper lacks a section that carefully details these points to enable reproducibility.

- The experiments and the environments considered seem impressive but should be better introduced. The authors might dedicate more space to explain what is the observation space, action space, and reward function. For instance, the Ant-Craft environment is very briefly introduced and it remains unclear to me what it really consists of.

- The authors compare their methods only to ablations. It would be important to consider baselines such as the ones mentioned in the previous points (AlphaNPI, Options method, …).

**Time Spent Reviewing:**

3

---

> ### Author Response · Authors · 2021-08-11
> **Answers to reviewer 1PbL's questions to help understanding**
>
> We answer the reviewer's questions here. We hope that this can help the reviewer gain a better understanding of our paper.
>
> > This paper was very difficult for me to follow. I think it would require more care to introduce the concepts, position the method with respect to the literature, and better detail the examples.
>
> We will try our best to polish our paper to make it accessible to readers with different backgrounds. Please see below for detailed answers to specific questions.
>
>
> > I did not get in the introduction what is precisely the problem that the authors aim to solve and why the current methods fail to do so.
>
> L1-6 in the Abstract, and L17-37 in the Introduction of our paper describes these.
>
> The problem we aim to solve is to develop a technique for automatically synthesizing the guiding programs for reinforcement learning policies under partial observations. The current program-guided approaches need the user to manually provide the guiding programs for every new task, which induces high user burden.
>
>
> > At the very beginning of the paper, both concepts Domain-specific language and Task-specific program remain unclear to me. Especially, the example “gather wood” is used for both.
>
> We refer to these concepts as they are the user-provided information in prior works on program guided RL:
> - Program Guided Agent, Shao-Hua Sun et.al., ICLR 2020
> - Modular multitask reinforcement learning with policy sketches, Jacob Andreas et.al., ICML 2017
> - A Composable Specification Language for Reinforcement Learning Tasks, Kishor Jothimurugan et.al., NeurIPS 2020
> - LTL2Action: Generalizing LTL Instructions for Multi-Task RL, Pashootan Vaezipoor et.al., ICML 2021
>
> Domain specific languages are a basic and fundamental concept in program synthesis. It is the language that the synthesized program is written in. Here are some example papers (from many of them) where the reviewer can find some more example DSLs that could help understanding:
> - LEVERAGING GRAMMAR AND REINFORCEMENT LEARNING FOR NEURAL PROGRAM SYNTHESIS, Bunel et.al. ICLR 2018
> - RobustFill: Neural Program Learning under Noisy I/O, Devlin et.al., ICML 2017
>
> Task-specific programs are the guiding program provided by the user to every new task. It is written in the DSL, so it is natural that “gather wood” appears in both, since it is a construct in the DSL.
>
>
> > The authors often mention that their method alleviates the user burden to provide a guiding program for every new task. What the authors mean here remains unclear to me and I believe an example would help to better understand.
>
> L31-37 in the Introduction explains this user burden in prior work. Section 2 provides an example of how our approach works. We will elaborate more here.
>
> Taking the 2D-Minecraft environment as an example. In prior works, for every new task (a map configuration and a goal), the user needs to manually write a guiding program. For example, Figure 1a in the paper shows a task (the map and the goal of getting a gem). In prior work, the user needs to manually write a program:
> ```
> get wood; use workbench; get iron; use factory; use axe; get gem
> ```
> to instruct the agent how to achieve this task. Our work let the agent automatically synthesize this program, without the user providing it.
>
>
> > Many concepts are mentioned without being introduced first. For instance, in the very beginning, the authors talk about options without explaining what options are.
>
> Option is a widely used concept in RL and planning. An option is a tuple $(\pi, \beta)$ that executes the policy $\pi$ until the termination condition $\beta$ holds. We introduce options in L149-151 in the Problem Formulation (Section 3).
>
>
> > Many mathematical notations are also not properly introduced. For instance, on page 2 in the definition of the executor, p = c 1 ; ...; c k, the variables c_1, …, c_k are not introduced and I do not understand what they refer to.
>
> $c_1, …, c_k$ are components of the program (see L101-104, L153-160).
>
>
> > Section 2 covers a motivating example that remains unclear to me. I think that this section is very abstract for a motivating example and that most of the mathematical notations might be kept for later in the paper and replaced here by concrete examples.
>
> > In the related work, the authors missed the literature about Neural Programmer Interpreters and notably AlphaNPI that is also an RL method guided by programs that would be relevant as a baseline in this study.
>
> Neural Programmer-Interpreters (NPI) is a technique for learning a neural network to execute programs. It is orthogonal to the problem we aim to tackle; we focus on how to automatically synthesize guiding programs for reinforcement learning tasks. AlphaNPI is using RL to train a NPI, rather than using NPIs for RL as the reviewer suggested. Therefore, it is not on the same topic of program guided RL which is what we are doing.
>
>
> > The authors mention several times options but do not cover them properly in the related nor compare to them. I think that at least the Option-Critic and HAC should be mentioned, and even compared to, in this study. I understand the point made by the authors that work over options does not cover the partially observed setting however, a simple baseline to validate that this method is more relevant than using options might be to consider an option algorithm such as HAC and add a recurrent core to the high-level policy to handle partial observability.
>
> First of all, Option-Critic and hierarchical option-critic (we assume this is what the reviewer refers to as HAC) focuses on automatically learning options, which is orthogonal to the problem we aim to tackle. Therefore they are not suitable baselines for our approach. A more related line of work is on planning with options. As we discussed in our related work section (L74-79), this line of work does not handle planning with options under partial observations.
>
>
> > The different neural networks architecture as well as the RL agent used to train MPPS remain unclear to me. I think that the paper lacks a section that carefully details these points to enable reproducibility.
>
> We discussed the neural network architectures in Sections 6.2 and 6.3. We discussed the RL algorithm for training in Section 5. We will try our best to discuss in more detail in the appendix.
>
>
> > The experiments and the environments considered seem impressive but should be better introduced. The authors might dedicate more space to explain what is the observation space, action space, and reward function. For instance, the Ant-Craft environment is very briefly introduced and it remains unclear to me what it really consists of.
>
> We discussed the experiments and environments in Section 6.1. We will try our best to provide more details in the appendix.
>
>
> > The authors compare their methods only to ablations. It would be important to consider baselines such as the ones mentioned in the previous points (AlphaNPI, Options method, …).
>
> We compared our approach to prior works as baselines, including world models, relational reinforcement learning, and program guided agent. Section 6.2 describes these baselines in more detail.

---

> > ### Comment · Reviewer_1PbL · 2021-08-31
> > **Thank you!**
> >
> > Thank you for your thorough and detailed response that answered most of my concerns. However, I still believe that this paper lacks clarity and would benefit from a rewriting of some parts and some polishing. Although I am not an expert in using programs to guide reinforcement learning algorithms, I have a good knowledge of deep reinforcement methods and still, I really struggle despite your clarifications to understand several parts of the paper. Given the scope of the paper, I think it should be also accessible for people less familiar with programs but that are familiar with deep reinforcement learning. Also, the paper lacks rigors in many places, such as in the formulations or in the mathematical formula. All in all, I strongly encourage the authors to keep working on this track and to clarify the redaction as this work line is really exciting!

---

> > > ### Author Response · Authors · 2021-08-31
> > > **Any specific questions? We are happy to answer!**
> > >
> > > Thank you for your reply. We are glad that our response answered most of your concerns. Could you be more specific on what are the parts that you still struggle to understand, and what are the parts that you find lacking rigor? We are happy to answer any remaining questions you have!

---

### Official Review · Reviewer_kENU · 2021-07-24

**Rating:** 4
**Confidence:** 3

**Summary:**

This paper proposes a neurosymbolic, hierarchical agent architecture which uses program synthesis at the top level of abstraction to generate a symbolic plan. The plan is executed using learned options, which are trained via supervision. To address partial observability, a generative model of symbolic states given observations is trained and used to sample possible states of the world, which the program synthesizer optimizes with respect to. Replanning (i.e. re-running the synthesizer) is performed every so often in order to update the plan/program with newly observed information.

I really like the idea of learning a generative model over states and performing program synthesis with respect to that. This idea has close ties to other approximately Bayes-optimal methods for handling partial observability, and seeing it successfully instantiated in the program synthesis setting is an important contribution. More generally, the topic of the paper is timely and important, as program synthesis and neurosymbolic methods more generally are an important area of current AI research.

However, my sense is that this paper is not ready for publication at NeurIPS. This is primarily due to (1) a lack of clarity and (2) inappropriate and poorly-tuned baselines; secondarily, I am also concerned by the (3) overreliance on privileged information, and (4) lack of grounding in the automated planning / TAMP literature (though to a lesser degree than points 1 and 2).

**Limitations And Societal Impact:**

I have discussed a number of limitations above, such as the assumption of full observability during training. There are a few other less important limitations as well. For example, environments must be static (i.e. once something is observed, it is assumed to not change unless it is within the agent’s field of view), and actions are deterministic. The proposed method would not work well if either of these assumptions were violated. I don’t necessarily expect the paper to address this, because it can’t be expected to solve all problems at once, but these limitations should be more explicitly called out.

**Main Review:**

**Clarity**

The paper suffers from a major lack of clarity. I had to re-read the methods multiple times before I felt like I understood what was really being done, and I’m still confused about many things. There are many details missing which make it hard to understand exactly what experiments were run, how they were evaluated, how the baselines are implemented, and so on. I list below some things that particularly confused me, but as a more general recommendation, I would suggest including much more information in the appendix and referring to it liberally. There should be enough information in the paper for someone to be able to reimplement it in entirety, and that is currently far from the case.

* What exactly is included in the observations that the hallucinator is trained on? The paper says “we assume the observation o on the current step already encodes all observations so far”. One part of the paper led me to believe that o was the count of state features (e.g. the number of gems), rather than the full map itself. But after a while longer of combing through the paper, now I’m back to thinking that observation is the more typical use of the word (i.e. the map). But how is this encoded? Is it an image? Is it a more symbolic representation? Also, what does “all observations so far” mean? Is that a matrix of observations from all previous timesteps? Or do you just mean you assume the Markov property here?
* It sounds like the hallucinator is trained using ground-truth state information, but this is not explicitly stated. Is this true? If not, how do you deal with the fact that a random policy is unlikely to be able to actually observe the full state (since for long-horizon tasks this will be exponentially unlikely)?
* It’s not clear to me if the hallucinator and executor are trained simultaneously or not, and it’s not clear what the x-axis of Figure 3 shows. Are those episodes required by the executor? Does it include the data required to train the hallucinator, too?
* There are almost no details given about the baselines, which makes it hard to evaluate whether they are appropriate. For example, with the End-to-End baseline, the paper states there is “one DNN policy per goal”. What does this mean? How are the separate policies trained? How many policies does that amount to?
* What are the goal specifications that the agent is evaluated on? In your results, are you averaging across different types of goals? Are these evaluated in equal proportion? On line 230, the paper says the goal can be “one of 10 possibilities” but as far as I can tell it is not stated anywhere what these possibilities are. It would help to include this type of detail in the appendix.
* Do all agents (including baselines) get to see the fully-observed environment during training? The paper states that “just for training [the executor], we use the ground truth program p synthesized based on the fully observed environment (since we can explore the entire map and post-hoc generate p)”. However, this seems to me to put the program synthesis agent at a very unfair advantage compared to the other agents, which presumably don’t get to see the fully-observed environment during training in any capacity. The other agents should either also get to see the fully-observed environment, or you need to explicitly include the data cost for exploring the entire map to generate p. (It’s also not clear to me that all maps can even be fully explored in a reasonable amount of time, given that part of the map may be blocked off by rocks/water).

**Quality**

There are two issues that pertain to quality: baselines and privileged information.

First, the baselines seem poorly chosen and poorly tuned. Given the main contribution of the paper is a method for dealing with partial observability via program synthesis, it would seem to me the more appropriate comparisons would be with either (1) other approaches to program synthesis that don’t account for partial observability or (2) neural approaches that do account for partial observability. However, the paper only compares to neural approaches that don’t deal with partial observability, which seems inappropriate.

But assuming the choice of baselines themselves is ok, I had a lot of confusion about how they were even implemented (see above), which makes it hard for me to evaluate whether they are fair. From what I was able to gather, I get the impression that they are not. For example, the paper says on Line 266 that “We use the same actor (resp., critic) network architecture 267 across all approaches—i.e., an MLP with 128 (resp., 32) hidden units”, but this seems like a very poor baseline: it is probably not deep enough (the number of layers is not stated, but I’m assuming just 1?), 32 hidden units is almost certainly too small for the critic, and the architecture should probably be a CNN, not an MLP, as MLPs typically do not perform very well in visual/grid-world settings. Similarly, for Box World, where a CNN is used, the paper states that only one 3x3 conv layer is used, which is also probably not the best choice. Why was a different architecture chosen than what was used in the RRL paper (which is two 2x2 convs followed by ReLU)? It is suspicious to me that the relational approach does *worse* on Box World than the other neural approaches, which is the reverse of what was found in the RRL paper. Including experiments in the appendix indicating that these baselines are correctly implemented (e.g., that they succeed at the fully-observed version of the task) would help me have more confidence that they aren’t a strawman comparison.

Second, the method relies on a lot of privileged information and domain knowledge. Specifically, from what I could tell, the method requires: a DSL, counts of state features (which may include information above and beyond the DSL, e.g. zones, boundaries, etc.), transition operators (referred to as “components” in the paper), a logical goal specification, and access to the fully-observed environment during training. I’m certainly willing to allow for some amount of this; I think it’s hard to make progress with program synthesis and other symbolic approaches without at least allowing access to a DSL (and in much of the TAMP literature, transition operators and goal specifications are also given). But the choice that pushes it over the threshold for me, at least, is to allow the agent to see the fully-observed environment during training, which really just seems like it defeats the purpose. Isn’t the whole point of the paper to have a learning algorithm which can deal with partial observability?

**Originality & Significance**

As far as I am aware, applying the Thompson sampling technique to program synthesis in order to account for partial observability hasn’t been explored. As I said previously, I think this is a great idea and it’s important to demonstrate how to get it to work in practice.

However, the paper tries to lay claim to another contribution which I don’t think is justified: ”the user only needs to provide a high-level specification for the goal of the task; then, the agent automatically synthesize a suitable program that is used to guide the policy” (Line 34). In particular, the method is not really doing program synthesis but rather task planning—which is an area that has been highly studied in the context of task and motion planning (TAMP). Like the proposed method, TAMP relies on having access to the DSL, goal specification, and transition operators. Yet, TAMP isn’t even mentioned in the paper at all. Perhaps the authors aren’t aware of this area of research, in which case, here is a recent review of such methods [1]. Moreover, while TAMP is not my area of expertise, from a quick literature search, I found a few papers that propose to deal with partial observability in TAMP [2, 3]. I think that TAMP at least deserves a mention in the related work section, and I think that the paper should tone down its claims about the program synthesis in and of itself being a novel contribution.

**Minor comments:**

* Typically when comparing learning curves of RL agents, the x-axis should report the number of environment transitions, not the number of episodes.
* What does the *1000 mean in Figure 3? Is that indicating that the x-axis actually goes from 0 to 200k or 400k episodes?
* “We leverage program synthesis [27] with the world models approach [9] to address these issues” → this statement isn’t really accurate, your generative model of states given observations isn’t really what world models does. I would recommend being more precise and just saying you train a generative model over states given observations.
* The approach of sampling a realization of an MDP and planning optimally with respect to it—i.e. essentially the Thompson sampling approach—is approximately Bayes optimal in some settings, e.g. see [4]. Your approach is quite similar and it might be worth discussing this.
* Figure 7 is very hard for me to interpret. I would suggest not having two scales on the same plot but instead plot the variance as error bars around the mean.
* “Significantly” is used a number of times throughout the paper to indicate a substantial effect. However, “significant” has a precise statistical meaning, which is that we can be confident one value is higher than another. Since you perform no statistical tests in the paper, I would recommend avoiding the use of the term “significant” and stick to more general terms like “much higher/lower” or “substantially”.

[1] Garrett, C. R., Chitnis, R., Holladay, R., Kim, B., Silver, T., Kaelbling, L. P., & Lozano-Pérez, T. (2021). Integrated task and motion planning. Annual review of control, robotics, and autonomous systems, 4, 265-293.
[2] Phiquepal, C., & Toussaint, M. (2019, May). Combined task and motion planning under partial observability: An optimization-based approach. In 2019 International Conference on Robotics and Automation (ICRA) (pp. 9000-9006). IEEE.
[3] Kaelbling, L. P., & Lozano-Pérez, T. (2013). Integrated task and motion planning in belief space. The International Journal of Robotics Research, 32(9-10), 1194-1227.
[4] Guez, Silver, & Dayan (2013) “Scalable and Efficient Bayes-Adaptive Reinforcement Learning Based on Monte-Carlo Tree Search”

**Time Spent Reviewing:**

4.5

---

> ### Author Response · Authors · 2021-08-11
> **Response (Part I)**
>
> We thank the reviewer for their detailed review. We are glad to see your appreciation of our motivation and idea. We hope the following response will help clarify your questions and address your concerns. We will add clarifications on these questions to our paper.
>
> ### Hallucinator observations
>
> The observation $o$ refers to the map with all the previously observed grid cells up to the current timestep. Since our environment is static, this map captures “all observations so far” (Markov property). This map is fed to the executor policy as inputs.
>
> For the hallucinator, we train it to operate on state features (e.g. the counts of gems). It takes the state features of the observation $o$ as inputs, and predicts the state features of the full map.
>
> ### Hallucinator trained on ground truth states
>
> The hallucinator is trained using the ground-truth state information. We will explicitly state this in the paper.
> Our work considers a multi-task setting, where we have a set of sampled training tasks (goal & world configurations) and a held-out set of testing tasks. The goal is to learn a policy using the training set that achieves good performance on the test set. In such settings, it is common practice to use ground truth information of the training set during training -- this is consistent with existing work [1,2]. This will not invalidate the findings since all the evaluation results (training curves, Table 1 etc.) are evaluated on the test set, where no ground-truth information is given.
>
> ### Training curve
>
> The hallucinator and the executor are trained separately. The x-axis of figure 3 refers to the episodes for executor training. We will update the plot to include the episodes for training the hallucinator. Note that the number of training episodes for the hallucinator are much lower than the number of training episodes for the executor:
>
> | Environment | #Episodes for hallucinator | #Episodes for executor | hallucinator/executor (%) |
> | ---------------- | ------------------ | ------------------- | ------------ |
> | 2D-craft | 200 | 400,000 | 0.05% |
> | Box-world | 1,000 | 200,000 | 0.5% |
>
> We can see that the number of training episodes for the hallucinator is only 0.05% of the number of training episodes for the executor in 2D-craft, and 0.5% in box-world. So adding the hallucinator training steps to the training curve will not substantially change the training curves.
>
> ### Baseline details
>
> The end-to-end baseline uses one DNN policy per type of goal. For 2D-craft, we have 10 types of goals for all tasks (e.g. get gem/build bridge) [L230]. The end-to-end baseline has one network for each of the goal types. These policies are trained together using the actor-critic algorithm with curriculum learning, same as our approach [L250]. So essentially the end-to-end baseline has 10 actors, one per goal type, and for each training task, the corresponding actor network is used; the executor in our approach has 11 actors, one per component, and the actor network corresponding to the current program component is used (so for each training task, multiple actors will potentially be used depending on the guiding program). Except for this difference, the way we train end-to-end baseline and the executors are exactly the same. Essentially our approach decomposes the policy into components based on the guiding program, whereas end-to-end baseline treats the policy for each type of goal as a whole and does not decompose it.
>
> ### Evaluations
>
> The 10 types of goals we evaluated on are: get wood, get iron, get grass, get gold, get gem, build plank, build stick, build bridge, build axe, build ladder. Our test set contains tasks from each of these 10 types. To make the test set more challenging, we include more (15 tasks) from the two hardest goals: get gold, get gem. These goals involve potentially longer horizons to achieve. The rest of the goals are in equal proportion. All our results are averaged over the test set, therefore averaged across different types of goals. This setup is also used in prior works [1,2]. We will add these details into the Appendix.
>
> ### Full map information during training
>
> We performed additional experiments that train the baselines (end to end, world model) with fully observed maps for 2D-craft. The performance at the end of training (400k episodes) are as follows:
>
> | Approach | Average Reward | Average Finishing Step |
> | ------------ | -------------- | ------------------- |
> | End-to-end | 0.14 | 87.6 |
> | World models | 0.14 | 88.0 |
> | Ours | 0.70 | 56.4 |
>
> These results show that the fully observed map information during training does not help the baselines learn faster or better, and is therefore not the reason behind the performance difference between our approach and the baselines.
>
> The reason we use programs synthesized from fully observed maps during training is to keep training simple; our approach can be adapted to interweave learning the executor and using program synthesis to generate programs based on partial observation, though doing so would be slower due to the need to run synthesis during training.
>
> We believe it is reasonable to assume access to fully observed information for the training environments. In particular, we are considering the multi-task setting, where we have a set of sampled training tasks (goal & world configurations) and a held-out set of testing tasks. The goal is to learn a policy using the training set that achieves good performance on the test set. In such settings, it is common practice to use ground truth information of the training set during training [1]. This will not invalidate the findings since all the evaluation results (training curves, Table 1 etc.) are evaluated on the test set, where no ground-truth information is given.
>
> ### Baseline choice
>
> We do compare with (1) other approaches to program synthesis that don’t account for partial observability, and (2) neural approaches that do account for partial observability.
>
> We have an optimistic synthesizer baseline that synthesizes the shortest possible program making best-case assumptions about the unobserved parts of the map (Figure 6, Figure 5 c, d). We also have a random hallucinator baseline that randomly samples the unobserved parts of the map instead of learning a conditional generative model (hallucinator). These two are the closest baselines we could think of with a program synthesis approach that does not account for (or naively account for) partial observability.
>
> We want to emphasize that our baseline neural approaches do handle partial observability. End-to-end, world models, and relational approaches can all handle partial observation as inputs. In particular, our world models baseline is particularly designed for partially observable environments.
>
> ### Baseline architecture
>
> First of all, the choice of MLP models and their sizes are from the prior work that originally introduced the 2D-craft environment [2]. We adopt the same model architecture as the above work for the actor and critic networks. Moreover, we use the same architecture for both our approach and the baselines, so our approach does not gain any unfair advantage.
>
> As for the choice of MLP instead of CNN, we follow the same preprocessing method on the inputs (maps) before feeding into the MLP policy network as in [2]. This preprocessing method extracts the 5x5 grid around the current position of the agent as the fine-scale features, and also an aggregated 5x5 grid of coarse-scale features which is aggregated over a 25x25 region from the original map via max pooling. Therefore, this can be viewed as a hand-written convolution and pooling operation before feeding into the MLP. More details can be found in [2] and its github code repository.
>
> **Relational:**
> We experimented with the original architecture from the DRRL paper (two 2x2 convs). Here is the performance at the end of training (200k episodes):
>
> | Architecture | Avg. Reward | Avg. Finish Step |
> | -- | -- | -- |
> | Original (two 2x2 convs) | 0.7 | 54.3 |
> | Our version (one 3x3 conv) | 0.75 | 53.7 |
>
> The DRRL paper did not release their code, so we do not have access to their implementation and the set of hyperparameters they use. We have experimented with different number of layers of the relational module, different number of heads, and different learning rate, but were yet to get the relational baseline to perform better than the end-to-end baseline. We want to point out that the DRRL paper does not compare with the same neural baselines as in our paper, which is another possible explanation.
>
> We feel the same way as the reviewer since the results are not as expected. But we decided to include these results in the paper since that is what the experimental results show.

---

> > ### Author Response · Authors · 2021-08-11
> > **Response (Part II)**
> >
> >
> > ### Privileged information
> >
> > First of all, we want to emphasize what is the main point of our paper and how we support this. The main goal of our paper is to develop an approach to automatically synthesize guiding programs for reinforcement learning policies under partial observations. Our proposed MPPS approach achieves this. All the evaluation results we report in the paper are on the test set where no ground truth/fully observed information is given to the agent. Before our approach, users need to manually provide the guiding programs in order to achieve good performance on these tasks. Our approach is able to automatically synthesize the guiding programs while maintaining the good performances, as is shown in the comparison with the oracle baseline in our experiments.
> >
> > Secondly, we believe it is reasonable to assume access to fully observed information for the training environments. In particular, we are considering the multi-task setting, where we have a set of sampled training tasks (goal & world configurations) and a held-out set of testing tasks. The goal is to learn a policy using the training set that achieves good performance on the test set. In such settings, it is common practice to use ground truth information of the training set during training [1].
> > This will not invalidate the findings since all the evaluation results (training curves, Table 1 etc.) are evaluated on the test set, where no ground-truth information is given.
> >
> > The reason we use programs synthesized from fully observed maps during training is to keep training simple; our approach can be adapted to interweave learning the executor and using program synthesis to plan, but doing so would be slower due to the need to run synthesis during training.
> >
> > Thirdly, about the required domain knowledge. As the reviewer points out, much of the existing work on program synthesis and other symbolic approaches also requires a DSL, transition operators and goal specifications. Furthermore, as with many existing works in planning and RL, we assume that the state abstraction is given [3, 4]. Automatically learning/generating these state abstractions is an important research area by its own, with a long-history of work [5, 6]. We see the problem of automatically learning state abstractions as an orthogonal and complementary research area to our work.
> >
> > ### Related work
> >
> > We will include the TAMP literature in our related work. As the reviewer noted, TAMP by itself does not handle partial observability. Below we note some of the key differences between our approach and the follow-up TAMP based approaches (that the reviewer pointed out) that can handle partial observability.
> >
> > [2] This paper tries to learn a full symbolic program to handle all possible cases --- this program tends to be very complex (with many branches) and hence, hard to learn. In contrast, our approach learns a simple straight line program that is most likely to solve the task and then replans if needed. Besides, [2] handles only discrete partial observations; we are not restricted by that constraint.
> >
> > [3] This paper proposes to perform planning in belief space, which is more similar to our strategy. However, they make the significantly stronger assumption that a structured representation of belief space is available; in particular, they assume a probability distribution over the abstract state space is provided. In general, such a distribution can be exponentially large in the number of abstract states (though they appear to assume the predicates are independent). Furthermore, most deep generative models are unable to explicitly provide the distribution over abstract states, either providing samples (e.g., GANs and VAEs) or probabilities of given states (e.g., Normalizing Flows and VAEs). Thus, it would be difficult to apply this approach to our environments, where such a probability distribution over the environment is difficult to obtain.
> >
> > Finally, we believe that our model predictive program synthesis approach is novel. It uses a hallucinator and MaxSAT synthesis to synthesize guiding programs that robustly account for multiple possible futures. We are not aware of any existing approach that achieves this.
> >
> > ### Minor comments
> >
> > > Typically when comparing learning curves of RL agents, the x-axis should report the number of environment transitions, not the number of episodes.
> >
> > We also have the training curves in terms of the number of environment steps. It shows the same trend as the training curve in terms of the number of episodes. We will include the training curves with the number of steps in the appendix.
> >
> > > What does the *1000 mean in Figure 3? Is that indicating that the x-axis actually goes from 0 to 200k or 400k episodes?
> >
> > Yes, this is correct.
> >
> > We thank the reviewer for the rest of the suggestions on the writings in the minor comments section, and will do our best to incorporate them into our paper.
> >
> > ### Limitations
> >
> > We will add a detailed limitations section in our paper.
> >
> > We do not anticipate major difficulty to apply our MPPS approach to environments where actions are non-deterministic. Our MaxSAT synthesis works with the logical formulae of the components that describe their *intended* behaviors. These logical formulae can be used even if the underlying environment is non-deterministic. Non-deterministic environments might increase the difficulty of the executor, since it must still learn to achieve the desired behavior in the face of a non-deterministic environment. But it will not affect the hallucinator and MaxSAT synthesizer, which are the main novelty of our approach.
> >
> > We hope that our response addresses the reviewer's questions and concerns. We are happy to discuss more if the reviewer has further questions.
> >
> > [1] Program Guided Agent, Shao-Hua Sun et.al., ICLR 2020
> >
> > [2] Modular multitask reinforcement learning with policy sketches, Jacob Andreas et.al., ICML 2017
> >
> > [3] Hierarchical Task and Motion Planning in the Now, Leslie P. Kaelbling et.al., ICRA 2011
> >
> > [4] Value Preserving State-Action Abstractions, David Abel et.al., AISTATS 2020
> >
> > [5] Skill Discovery in Continuous Reinforcement Learning Domains using Skill Chaining, George Konidaris, Andrew Barto, NIPS 2009
> >
> > [6] Automatically Generating Abstractions for Planning, Craig A. Knoblock, Artificial Intelligence, 1994

---

> > > ### Comment · Reviewer_kENU · 2021-08-16
> > > **A few more questions**
> > >
> > > Thank you very much for your clarifications to all my questions. I stand corrected on a few points (e.g. I agree that your optimistic and random baselines are good choices for comparing against program synthesis approaches that don't account for partial observability---though this could probably be made clearer in the paper).
> > >
> > > I have a couple more questions I was hoping you could answer:
> > >
> > > 1. Thank you for reporting the performance at the end of training given fully-observed inputs for the baselines. For the numbers you report above, are those on the test environments (i.e. partially observed) or on the training environments (fully observed)? If they are on the test environments, could you provide numbers for the performance on the training environments themselves as well? (The reason I'd like to see these is to better understand how strong the baselines are---e.g. if they are doing poorly on the fully-observed training environment, then they are probably not tuned well enough, and therefore may not be a fair comparison. But if they are solving the fully-observed training tasks, then I would be much more confident they are strong baselines!).
> > >
> > > 2. Standard, non-recurrent approaches to deep RL don't handle partial observability well unless they filter past information into a belief state (e.g. using an RNN) and even then, RNN-based approaches typically only handle minor forms of partial observability. To be more precise, partial observability is a hard problem in RL because capturing epistemic uncertainty (i.e., what you could know, but don't) is very challenging, as opposed to aleatoric uncertainty (i.e., statistical variation in the observations). MPPS is a neat approach for dealing with epistemic uncertainty. However, as far as I understand them, none of the baselines in the paper are designed to handle epistemic uncertainty (World Models does use a recurrent policy with a stochastic generative model, but that only captures variation in the aleatoric sense, not epistemic). If I am wrong about this, could you please clarify how the baselines handle epistemic uncertainty?
> > >
> > > (In my mind, I'm thinking that a good neural baseline would look something like this: train a hallucinator as in MPPS, but to predict the map directly rather than feature counts. Then train the neural policy in the fully-observed environment. Then at test time, sample N predictions from the generative model, use them to get N actions from the policy, and choose the action that shows up the most (or if in a continuous space, fit a density model and use the mean). This would be a good baseline because it holds constant the approach to partial observability, as well as the access to full-observability during training, while varying only along the neural vs. symbolic dimension).

---

> > > > ### Author Response · Authors · 2021-08-18
> > > > **Re: A few more questions**
> > > >
> > > > Thank you for your reply and suggestions! Here are the answers to the additional questions:
> > > >
> > > > > Thank you for reporting the performance at the end of training given fully-observed inputs for the baselines. For the numbers you report above, are those on the test environments (i.e. partially observed) or on the training environments (fully observed)? If they are on the test environments, could you provide numbers for the performance on the training environments themselves as well? (The reason I'd like to see these is to better understand how strong the baselines are---e.g. if they are doing poorly on the fully-observed training environment, then they are probably not tuned well enough, and therefore may not be a fair comparison. But if they are solving the fully-observed training tasks, then I would be much more confident they are strong baselines!).
> > > >
> > > > The results we reported above are on the test environments (partially observed). We performed additional evaluation of the baselines (end-to-end, world models) on the fully observed version of the test set (i.e. both training and testing are on fully observed environments). The results are as follows:
> > > >
> > > > | Approach | Average Reward | Average Finishing Step |
> > > > | ------------ | -------------- | ------------------- |
> > > > | End-to-end | 0.24 | 79.7 |
> > > > | World models | 0.24 | 79.2 |
> > > > | Ours | 0.70 | 56.4 |
> > > >
> > > > We can see that there is still a large performance gap between non-program-guided baselines and program-guided approach. This is consistent with the results reported in prior work: [Modular Multitask Reinforcement Learning with Policy Sketches, Jacob Andreas et.al., ICML 2017](https://arxiv.org/pdf/1611.01796.pdf) experimented with a simpler version of 2D-craft (8$\times$8 map, without zone structure), fully observable, and showed that program guided approach significantly outperforms non-program-guided approaches (see Figure 4 in their paper). The reason is that 2D-craft contains challenging tasks that involve a long sequence of subgoals, e.g. to build an axe, the agent needs to 1) get wood, 2) use workbench to build stick from wood, 3) get iron, 4) use factory to build axe from stick and iron. Without program guidance, the agent needs to learn this whole sequence from scratch in order to learn the policy for building axe. This is a very hard search problem for RL. Programs provide the structure of this sequence directly, therefore decomposing the policy learning into smaller sub-problems. This makes the search problem for RL significantly simpler.
> > > >
> > > > To sum up, the performance difference between non-program-guided baselines and our approach is due to program guidance; prior work has shown that non-program-guided approaches cannot solve the fully observed version of 2D-craft well. We see similar results in our experiments, and it is not due to poorly tuned baselines.
> > > >
> > > > > Standard, non-recurrent approaches to deep RL don't handle partial observability well unless they filter past information into a belief state (e.g. using an RNN) and even then, RNN-based approaches typically only handle minor forms of partial observability. To be more precise, partial observability is a hard problem in RL because capturing epistemic uncertainty (i.e., what you could know, but don't) is very challenging, as opposed to aleatoric uncertainty (i.e., statistical variation in the observations). MPPS is a neat approach for dealing with epistemic uncertainty. However, as far as I understand them, none of the baselines in the paper are designed to handle epistemic uncertainty (World Models does use a recurrent policy with a stochastic generative model, but that only captures variation in the aleatoric sense, not epistemic). If I am wrong about this, could you please clarify how the baselines handle epistemic uncertainty?
> > > >
> > > > The recurrent part of the world models approach, i.e. the MDN-RNN (M) Model, is trained to predict future observations conditioned on past observations. This objective encourages the M model to learn epistemic uncertainty. For example in the context of 2D-craft, in order to predict future observations well, the M model needs to learn what is more likely in the unobserved parts of the map (i.e. the distribution of the unobserved parts of the map). Therefore, the world models baseline learns to capture epistemic uncertainty.
> > > >
> > > > > (In my mind, I'm thinking that a good neural baseline would look something like this: train a hallucinator as in MPPS, but to predict the map directly rather than feature counts. Then train the neural policy in the fully-observed environment. Then at test time, sample N predictions from the generative model, use them to get N actions from the policy, and choose the action that shows up the most (or if in a continuous space, fit a density model and use the mean). This would be a good baseline because it holds constant the approach to partial observability, as well as the access to full-observability during training, while varying only along the neural vs. symbolic dimension).
> > > >
> > > > Thank you for this insightful suggestion! We agree this is a great baseline for comparing along the axis of program-guided versus non-program-guided.
> > > >
> > > > We hope that these answers address your questions. Please let us know if you have any further questions! Thank you again for the detailed comments and suggestions!

---

> > > > > ### Comment · Reviewer_kENU · 2021-09-01
> > > > > **Thank you for the further details**
> > > > >
> > > > > Thank you for providing the further details! This does make me more confident that the baselines are good. However, I don't feel that all my concerns were addressed. To summarize where I stand on each of the concerns from my original review:
> > > > >
> > > > > 1. Clarity - the clarifications you have provided in the rebuttal help a lot, but really all of this information should have been in the paper to begin with. My feeling is that to include so many details would really require a lot of rewriting, and thus warrant a complete re-review of the paper.
> > > > >
> > > > > 2. Baselines - I'm now satisfied with the program synthesis baselines that don't handle partial observability, and that the neural baselines are mostly reasonable. [ I still disagree that World Models is designed to handle true partial observability, though. World Models predicts a single (small) time step into the future, which will only capture small forms of aleatoric uncertainty (e.g. when revealing a previously hidden grid cell, maybe there is an item there, or maybe there isn't). It does *not* capture full belief states over the entire state of the world such as the full map, which is closer to what your generative model captures (e.g. whether there is a diamond anywhere on the map, or not). ]
> > > > >
> > > > > 3. Privileged information - I still think that the assumption of full observability during training is an unusual one in RL, and is a pretty big assumption to make (I suspect a random agent would not be able to reveal all the information in the training scenes). Even if a few papers in the past have made this, by and large it's an uncommon assumption in the RL literature.
> > > > >
> > > > > 4. Discussion of previous literature - My main point here was that the paper tries to lay claim to a contribution which I don’t think is justified: ”the user only needs to provide a high-level specification for the goal of the task; then, the agent automatically synthesize a suitable program that is used to guide the policy”. I understand that this claim comes from the fact that, in comparison to specific previous works like Program-Guided Agent, the program had to be provided directly, whereas that's not required in this paper. However, when taking a broader view of the literature, it's clear this isn't really a novel contribution: (1) first, it's not really program synthesis (it's task planning) and (2) there are entire fields (automated planning, task planning, TAMP) dedicated to this. So this is not something that can be claimed as a novelty, which is how it comes across in the paper. [ As a more minor point, using MaxSAT also doesn't appear particularly novel, based on a quick Google Scholar search ("maxsat task planning" or "sat task planning"), though perhaps your specific formulation of the problem is novel (this isn't my area of expertise, and I wouldn't be able to evaluate it). ] I don't think this is necessarily a problem for the paper: the approach to partial observability in a symbolic setting is, I think, a worthwhile contribution on its own. However, the paper needs to be much clearer about what is and is not a contribution (tying back to my other concern about clarity), and should be more grounded in the existing literature.
> > > > >
> > > > > I do think the ideas here are good, and there are some nice results, but these are diminished by the lack of clarity, assumptions of privileged information, and lack of grounding in the larger literature. I am therefore keeping my score the same. I definitely encourage the authors to continue working on this line of work though---it's very interesting, just not quite ready for NeurIPS!

---

> > > > > > ### Author Response · Authors · 2021-09-01
> > > > > > **Response to your remaining concerns**
> > > > > >
> > > > > > We are glad that our clarifications helped addressing your questions, as well as making you more confident about the baselines. Here are the answers to your remaining concerns.
> > > > > >
> > > > > > # Novel contribution
> > > > > >
> > > > > > First of all, we would like to emphasize that our main contribution is proposing a method that can automatically synthesize guiding programs under partially observable environments. This is our Model Predictive Program Synthesis (MPPS) approach.
> > > > > >
> > > > > > > (1) first, it's not really program synthesis (it's task planning)
> > > > > >
> > > > > > In terms of the problem we aim to solve, the guiding program (or plan) is a straight-line program, therefore the problem of automatically synthesizing such guiding programs is a program synthesis task. Synthesizing straight-line programs is a common task for prior work on program synthesis, e.g.:
> > > > > >
> > > > > > - Automatic Program Synthesis of Long Programs with a Learned Garbage Collector, Amit Zohar and Lior Wolf, NeurIPS 2018
> > > > > > - Write, Execute, Assess: Program Synthesis with a REPL, Kevin Ellis et.al. NeurIPS 2019
> > > > > >
> > > > > > > and (2) there are entire fields (automated planning, task planning, TAMP) dedicated to this.
> > > > > >
> > > > > > As we mentioned in our earlier responses, our work differs from prior work on planning in that we propose a novel approach that can synthesize guiding programs (or plans) *robustly under partially observed environments*.
> > > > > >
> > > > > > In comparison with the TAMP literature, as the reviewer noted in the review earlier, TAMP by itself does not handle partial observability. See our earlier response (Response (Part II)) for some of the key differences between our approach and the follow-up TAMP based approaches that the reviewer pointed out.
> > > > > >
> > > > > > Finally, we believe that our model predictive program synthesis approach is novel. It uses a hallucinator and MaxSAT synthesis to synthesize guiding programs that robustly account for multiple possible futures. We are not aware of any existing approach that achieves this.
> > > > > >
> > > > > > We are happy to discuss the relationship with any other prior work that the reviewer notices.
> > > > > >
> > > > > > > As a more minor point, using MaxSAT also doesn't appear particularly novel, based on a quick Google Scholar search ("maxsat task planning" or "sat task planning"), though perhaps your specific formulation of the problem is novel (this isn't my area of expertise, and I wouldn't be able to evaluate it).
> > > > > >
> > > > > > Existing work on using MaxSAT in planning either focuses on the problem of preference-based planning, where the goal is to synthesize a plan that maximizes the satisfaction of other preferred properties of the plan:
> > > > > > - Preference-Based Planning via MaxSAT, Farah Juma et.al., Canadian Conference on Artificial Intelligence 2012
> > > > > >
> > > > > > or uses MaxSAT to compute admissible heuristics for planning:
> > > > > > - MAXSAT Heuristics for Cost Optimal Planning, Lei Zhang and Fahiem Bacchus, AAAI 2012
> > > > > > - Optimal Partial-Order Plan Relaxation via MaxSAT, Christian Muise et.al., Journal of Artificial Intelligence Research 2016
> > > > > >
> > > > > > None of these works target the problem of planning under partial observation. Therefore, we believe our approach of using MaxSAT for robustly synthesizing guiding programs under partially observed environments is novel.
> > > > > >
> > > > > > > I don't think this is necessarily a problem for the paper: the approach to partial observability in a symbolic setting is, I think, a worthwhile contribution on its own. However, the paper needs to be much clearer about what is and is not a contribution (tying back to my other concern about clarity), and should be more grounded in the existing literature.
> > > > > >
> > > > > > We thank the reviewer for the suggestions. We will emphasize more in the paper that our contribution is synthesizing guiding programs under partial observations. We will include the discussions on the TAMP literature as pointed by the reviewer.
> > > > > >
> > > > > > # Baselines
> > > > > >
> > > > > > > I'm now satisfied with the program synthesis baselines that don't handle partial observability, and that the neural baselines are mostly reasonable.
> > > > > >
> > > > > > We are glad that the reviewer is now satisfied that the baselines are mostly reasonable.
> > > > > >
> > > > > > > [ I still disagree that World Models is designed to handle true partial observability, though. World Models predicts a single (small) time step into the future, which will only capture small forms of aleatoric uncertainty (e.g. when revealing a previously hidden grid cell, maybe there is an item there, or maybe there isn't). It does not capture full belief states over the entire state of the world such as the full map, which is closer to what your generative model captures (e.g. whether there is a diamond anywhere on the map, or not). ]
> > > > > >
> > > > > > In our tasks, when the agent takes one step, it can observe up to 5 hidden grid cells in the 2D-craft environment, and up to 7 hidden cells in the box-world environment. In order for the world models to predict the contents in these newly observed cells, it needs to capture some amount of epistemic uncertainty: what it could know (the contents in the newly observed cells), but don't (these are unobserved before). Because the observations in our environments are noise-free, we do not have aleatoric uncertainty in our environments.
> > > > > >
> > > > > > On comparing what distributions are learned by the world models and our approach, we would like to point out that they are actually equivalent. Denoting the observed parts of the map as $\textbf{x}$, and the unobserved parts of the map as $\textbf{y}=\{y_1, \dots, y_m\}$, where each $y_i$ is an unobserved cell. Our approach learns a joint distribution over $\textbf{y}$: $P(\textbf{y}|\textbf{x})$. World models learn a conditional distribution over $\textbf{y}$: $P(y_1|\textbf{x})$, $P(y_2|y_1, \textbf{x})$, etc.. Note that $P(\textbf{y}|\textbf{x}) = P(y_1|\textbf{x}) \times P(y_2|y_1, \textbf{x}) \times \dots$, so the two distributions are equivalent (you can compute one given the other). Therefore, fundamentally world models and our approach are both learning the same epistemic uncertainty.
> > > > > >
> > > > > > # Privileged information
> > > > > >
> > > > > > > I still think that the assumption of full observability during training is an unusual one in RL, and is a pretty big assumption to make (I suspect a random agent would not be able to reveal all the information in the training scenes). Even if a few papers in the past have made this, by and large it's an uncommon assumption in the RL literature.
> > > > > >
> > > > > > We would like to emphasize that we are considering the multi-task setting, where we have a set of sampled training tasks (goal & world configurations) and a held-out set of testing tasks. The goal is to learn a policy using the training set that achieves good performance on the test set. This settings is different from the settings where there is only one task (no separate train/test) and the agent learns to solve this task -- this is probably the more commonly studied settings in RL, e.g. inverted pendulum, car racing, VizDoom, Go etc.. We agree that for these single task domains, it is unusual to assume to know the fully observed map during training. But for the multitask settings we consider, it is common practice to use ground truth information of the training set during training (see our earlier response for references). The rationale is the same as supervised learning, where we have ground truth labels for the training data.
> > > > > >
> > > > > > # Clarity
> > > > > >
> > > > > > > the clarifications you have provided in the rebuttal help a lot, but really all of this information should have been in the paper to begin with. My feeling is that to include so many details would really require a lot of rewriting, and thus warrant a complete re-review of the paper.
> > > > > >
> > > > > > We are glad that the clarifications we provided helped a lot. We think the questions raised by the reviewers are very helpful, and we will include the contents of the discussions in our paper. In terms of the structure of our edits based on your review, we will:
> > > > > > - Add a dedicated section in problem formulation to describe the multitask settings we use (contents from this response, also in Response (Part I))
> > > > > > - Including more details on the hallucinator observations and training in section 4 and 5 respectively (contents from Response (Part I))
> > > > > > - Update the training curves to including episodes for hallucinator training (from Response (Part I))
> > > > > > - Include details about baselines and evaluations in the appendix (from Response (Part I))
> > > > > > - Include discussion of TAMP literature in the related work (from Response (Part II))
> > > > > >
> > > > > > These are the contents and structures of our rewrites. We would love to hear any further suggestions from the reviewer on the rewrites.
> > > > > >
> > > > > > We hope that our response addresses your specific concerns. We are happy to answer any additional questions. If the major remaining concern by the reviewer is on incorporating the discussions in the rebuttal into our paper, we acknowledge that we cannot fully address it right now since we cannot upload an updated version of the paper. But we hope that the contents of our rebuttal and the structure of rewrites above would help addressing this concern.

---

### Official Review · Reviewer_EBwR · 2021-07-27

**Rating:** 5
**Confidence:** 3

**Summary:**

Authors address a class of RL challenges where the main task implicitly consists of a sequence of subtasks, and the main goal of an agent can be split into two parts: to decompose the tasks into a sequence of subtasks (planner module) and to perform each subtask individually. They consider a scenario where the current state (map of the world) is not fully observable. The key idea of this paper is to use a model of the map to supply a planner module with a hypothetical map of the world so that it can devise an on-average well-working plan. The authors applied their method to several environments including a 2D Minecraft-inspired environment and show that it is able to outperform prior methods. Their method eliminates the need for the human user to produce a demo solution of the ultimate goal but still requires the user to specify the logical predicates of the ultimate goal and goals of micro-tasks that serve as milestones in planning. These predicates are based on some DSL also specified by the user.

Although this paper has some weaknesses (see below), it makes a step towards a promising direction, so I slightly lean towards accepting.

**Limitations And Societal Impact:**

I do not see any potential negative societal impact

**Main Review:**

Overview.
_____

The proposed three-piece approach helps RL agents in goal-oriented partially observed environments to complete complex multi-step tasks. Unlike other methods, which require that solution strategies are supplied, the approach needs only the specifications of sub-goals and the ultimate goal from the user.

A quite peculiarly named `hallucinator` piece is a conditional generative world model (C-VAE) trained on random exploration data, to predict realizations of unobserved sectors of the world map (in terms of their features) and hallucinate objects located therein, given the observations made so far.

The program `synthesizer` piece samples possible worlds conditional on the current observation history and applies a MaxSAT solver to compose a shortest program, i.e. a series of "components" (milestone micro-tasks) that guides the agent to the ultimate goal.

The final piece is the "executor", which traces the synthesized program by executing RL options associated with each "component" micro-task. The hallucinatory program synthesis step is repeated, should the agent fail to complete the goal or halt on running out of the allotted time, this time, however, leveraging newly collected observations.

Each option, i.e. a policy with a particular termination criterion, is pre-trained in the environment to complete the user-specified micro-task. For speedup, the rollouts during option pre-training are collected by leveraging the "ground truth program synthesized based on the fully observed [version of the] environment" (L217) for the ultimate goal.

Strong points.
_____

Like its predecessors Sun et al. (2020) and Andreas et al. (2017), the method produces a nice inductive hierarchical structure of goals and subgoals. A policy option is learnt for each specified zero-tier micro-task in the environment, and then the program synthesizer creates a script for a specified higher-tier task by composing lower-tier tasks and packaging this into an associated composite option.

This paper's method itself completes an important `micro-task` on the path to program-guided exploration, preferably synthesized on-the-fly, which would be capable of generating milestones from prior knowledge and micro-tasks, or a curiosity-based point of Interest detection method, that produces such micro-tasks for which the options are also learnt on-the-go.

Weak points / criticism.
______


1)	Experiments are set on some tasks composed especially the particular paper, and this is not obvious what should be SOTA for them. Also, five replications of each experiment seems to be on the side of being too few.

2)	Exploration: the proposed method of exploration is exploitative (goal-oriented) and do not have any curiosity features. Such approach seems to be limited to the environments with finitely explorable map. It is unlikely that their method would be successful for such challenges like obstacle tower challenge (Unity Technologies).

3)	Some important details of the paper are not clear:
- What maps S (should be) used to sample training data for a VAE hallucinator? Can we make sure that the (information from the) maps where the agents are tested do not leak to that training data for VAE?
- A connected question is: what does mean “the actual distribution P(s\mid o)”, how can it be defined and how can we sample it?
- What is “a randomly generated map” (L99). What is a generative model?

4) Although the proposed program synthesis approach is interesting, I am concerned that the variability and feature simplicity of the 2d-craft environment have played a significant role in the success of the presented experimental results. Indeed, the pre-trained micro-task options could memoize the peculiarities of this procedurally generated environment.

5) Another limitation of the particular instance of the presented method is the need for the full state to be accessible (or relevant features thereof). There are environments, e.g. the [NetHack](https://arxiv.org/abs/2006.13760), which have feature-rich observation space and very long branching exploration paths, and the full state of the “world” is unavailable, and has to be discovered. In this situation the hallucinator, which is trained on random exploration, will be extremely unlikely to ever stumble upon the ultimate goal, let alone be able to reconstruct the entire state. It will face the same challenge with feature-rich states, so there is a need to devise a method to limit the scope of the generative model to only the likely relevant locations.


Corrections, typos and RFCs
_______

- L291: It is unclear what is meant by “random hallucinator. Is it untrained, is not conditioned on the observations at all, or not updated with new exploration information?
- L290: It is also unclear what is meant by a synthesizer that makes “the best-case assumptions on about the unobserved parts” on?
- L107: “final state s_{-}” should be “final state s_{+}”?
- L 153: the conjunction sign should be rather conjunction?
- L35: “the agent automatically [sythesizeS]”
- according to NeurIPS style the captions should precede the tables.
- figure 6 should be table 2.
- increasing DPI on the included figures would enhance their accessibility


__________
After rebuttal

Summing up, in order to automatically find a program, which achieves the main goal of the agent, the MPPS method requires that we

* know the predicate specifications of micro-tasks (options) and the main goal;
* have well pre-trained policies that can explore well and complete micro-goals (options);
* have a generative model that reconstructs abstract state descriptions from partially observed data

The main contribution is an approach to automatically synthesize guiding programs for reinforcement learning policies under partial observability.

Having read all discussion, I slightly lower my score (6 -> 5). On one hand, the idea is promising, the direction seems important, and the authors addressed many important questions. On the other hand, I agree with kENU that many important details regarding training, ablations, and baselines are needed in the paper. Further, as detailed below, I think the paper needs additional experiments. The proposal about “interweaved hallucinator training with executor learning” should be discussed in the paper.

Now, although I am convinced that pre-training the component options under full observability is a way too privileged in the partially observed settings, I believe that such privileged setting still can give valuable proof-of-concept results for the core method: program synthesis with a help of a generative model for states. The availability of well-trained options can be considered as an assumption for the applicability of the method, and I believe that this assumption is reasonable for future RL systems, which will share options among tasks and environments.

However, I have additional concerns about baselines. Currently, the End-to-end baseline, the one with many micro-actor DNNs, is severely handicapped: (1) it does not have access to a hallucinator (mentioned in the response to Wt3P, `experiments/craft_nn.yaml#L7`, and `models/simple_nn.py#L59-66`), and (2) it is not recurrent (judging from a quick regex over the supplied codebase), contrary to the provided response to kENU. Point (2) is rather concerning since it is unclear how the non-markovian dependence on the full history is handled.

At the same time, the World Model baseline (recurrent) does not have access to pre-trained options, which makes the comparison unfair. Thus, the baselines do not allow us to separate the effect of good micro-goal options from the synergy between the hallucinator and the synthesizer.

I want to reiterate: in the codebase the ”hallucinator” is a CVAE with a Bernoulli decoder (`models/cvae.py#L122-124` and `train_cvae.py#L198`) trained on random rollouts (`train_cvae.py#L418-420`) to reconstruct full one-hot encoded (`train_cvae.py#L395-411`) abstract states (`environment/box.py#L444-470` and `environment/craft.py#L953-990`) from partial observations overlain on an empty binary grid map (`environment/box.py#L542-615` and `environment/craft.py#L953-990`).

In light of the claimed benefits of synthesized programs, I think a good baseline/ablation experiment would be to replace the synthesizer with a recurrent neural policy on pre-trained options, which __also has access__ to the hallucinator or an equivalent world model (also pre-trained on a fully observable environment). To fairly judge the benefit of program synthesis here it may be necessary to compare the number of requests each approach makes to the world model.

As for the claims of reduced user supervision, I suspect that identifying micro-goals and specifying them as logical predicates still demands that the user thoroughly analyzes the environment (L32). So the toning down of the contributions of the paper seems appropriate. Besides, micro-goals' logical predicates as they are described in the paper do not seem to be very high-level (L35): one has to tinker with the before- and after- states, enumerate positions, and zones in the grid, and compose conjunctive forms (L496). Despite this, I am inclined to agree that such specifications are more basic than a program description of an optimal solution as in Sun et al. (2017).

**Time Spent Reviewing:**

9

---

> ### Author Response · Authors · 2021-08-11
> **Thank you!**
>
> We are extremely thankful that the reviewer spent much of their valuable time to provide a deep and thoughtful review. The insightful and constructive suggestions are very valuable to us. Thanks a lot! Here we provide answers to the questions raised by the reviewer. Hopefully this can address the reviewer's concerns.
>
> ### 1. Experiments
>
> We would like to point out that the 2D-craft environment was originally introduced in [1], and was later used in other works as well, e.g. [2,3]. The box-world environment was introduced in [4].
>
> Prior work [1,2] has shown that methods with program guidance significantly outperforms non-program-guided approaches on the 2D-craft environment, achieving SOTA results. However, these program guided methods require user-provided programs for every new task. Our proposed method removes this user burden while maintaining the benefit of program guidance. The oracle baseline in our paper can be seen as the SOTA approach (method with user-provided guiding programs as in [2]). Our approach achieves a similar performance as the oracle, without requiring the user-provided programs. We will add this clarification to our paper.
>
> ### 2. Exploration
>
> Acting optimistically with respect to the predicted environment is known to be a good exploration strategy for reinforcement learning (i.e., “optimism in the face of uncertainty”, including the RMax and UCRL algorithms). Our algorithm can be seen as an instance of the posterior sampling strategy for exploration, which is also known to be effective for finite MDPs:
> - Osband et al., (More) Efficient Reinforcement Learning via Posterior Sampling. NeurIPS 2013.
>
> Furthermore, in our approach, the user has control over what components/options to define. For our current environments, we found the existing options already demonstrating effective exploration. For instance, an option such as “find wood” encourages the agent to explore the environment to achieve its goal. Our executor thus learns to balance exploration and exploitation to achieve this goal as quickly as possible. Generally speaking, for options in partially observed environments, the agent must learn to explore to achieve its goal.
>
> We agree with the reviewer that in certain more complex environments, other exploration strategies might be needed. We think studying different exploration strategies under our framework is an interesting future research direction.
>
> ### 3. Details on hallucinator training
>
> We consider the same multi-task setup as used in prior work [1, 2]. In this setup, we have a set of sampled training tasks (goal & world configurations) and a held-out set of testing tasks. All the tasks are sampled from a task distribution (over goal and world configurations). In our work and also the above prior work, this task distribution is manually written since the domain is a simulated game. For more real-world domains, this distribution can come from the real world.
>
> The maps used to sample training data for the hallucinator are from the sampled training maps. The test set is sampled separately from the training set, therefore simply memorizing the training maps would not help. Nevertheless, the distribution of the training maps does provide information on the distribution of the test maps -- this is a fundamental assumption of statistical learning.
>
> “The actual distribution $P(s\mid o)$” is induced by the underlying task distribution as mentioned above. We estimate it by fitting the conditional generative model on the empirical distribution from the training set.
>
> We thank the reviewer for raising these important clarification questions. We will add this clarification to our paper.
>
> ### 4. Difficulty of the environments
>
> First, we would like to point out that much of the existing work on partially observed environments only experiments on symbolic, grid-world environments, e.g. [5, 6, 7].
>
> In addition, the 2D-craft environment we consider is already a harder version than what is used in previous work: [1] uses 8x8 maps, fully observable, and no zone structure; [2] uses 8x8 maps, fully observable, with zone structures; our work uses 10x10 maps, partially observable, with zone structures.
>
> Moreover, our experiments show that non-program-guided baselines do not perform well on 2D-craft. This demonstrates that our tasks are challenging. The non-program-guided baselines are equally capable of memorizing the training maps. Should this be the reason behind the success of our approach, the non-program-guided baselines would also perform well on the tasks. Therefore, this is not the reason behind the success of our approach.
>
> ### 5. Availability of the full state
>
> We agree with the reviewer that in some environments with very long branching exploration paths, training the hallucinator with random exploration will not likely to cover all the observation space for estimating $P(s|o)$. This does not have any substantial effect on the environments we experiment on, and we believe it is also valid for many settings on robots operating in realistic environments (e.g., in a home or office building).
>
> For potential domains where this might be an issue (e.g. NetHack), we can interweave hallucinator training with executor learning and program synthesis, so that we finetune/retrain the hallucinator as the policy gets better. We think this is an interesting research direction. We will add a discussion for future work.
>
> ### Corrections, typos and RFCs
>
> > L291: It is unclear what is meant by “random hallucinator. Is it untrained, is not conditioned on the observations at all, or not updated with new exploration information?
>
> The random hallucinator is still conditioned on the observation so far, but untrained. It randomly predicts the configuration of the unobserved parts of the world. In our experiments, the hallucinator directly predicts the abstract state features, so the random hallucinator simply predicts random values for each entry of the state features (e.g. number of wood in zone 1) under the condition that it does not conflict with existing observations (e.g. predicting number of wood in zone 1 to be 1 when there are already 2 woods observed in zone 1). The purpose of this ablation is to demonstrate the importance of using a learned hallucinator. We thank the reviewer for raising this question, and will add clarifications on this in our paper.
>
> > L290: It is also unclear what is meant by a synthesizer that makes “the best-case assumptions on about the unobserved parts” on?
>
> It means that it considers the unobserved parts of the world to be in any possible configuration. If a program can achieve the goal under any one of these configurations, this program is considered to be correct. The optimistic synthesizer chooses the shortest program considered to be correct in this optimistic sense. For example, if the goal of the task is “get gem”, and there is some unobserved grid cells in the current zone, then an optimistic synthesizer will always synthesize the simplest program “get gem”, whereas in our approach the MPPS synthesizer might synthesize a program that first build a bridge to cross the water and then get gem, if that is considered more likely. This baseline also demonstrates the importance of using a hallucinator, instead of a heuristic such as pure optimism. We will add clarifications on this in our paper.
>
> > - L107: “final state s_{-}” should be “final state s_{+}”?
> > - L 153: the conjunction sign should be rather conjunction?
> > - L35: “the agent automatically [sythesizeS]”
> > - according to NeurIPS style the captions should precede the tables.
> > - figure 6 should be table 2.
> > - increasing DPI on the included figures would enhance their accessibility
>
> We deeply appreciate the reviewer for the careful reading and their suggestions. We will make these edits in our paper.
>
> Again, we thank the reviewer for the constructive suggestions! We hope that we have addressed the reviewer's concerns. We are happy to answer any further questions!
>
> [1] Modular multitask reinforcement learning with policy sketches, Jacob Andreas et.al., ICML 2017
>
> [2] Program Guided Agent, Shao-Hua Sun et.al., ICLR 2020
>
> [3] Using Reward Machines for High-Level Task Specification and Decomposition in Reinforcement Learning, Rodrigo Toro Icarte et.al., ICML 2018
>
> [4] DEEP REINFORCEMENT LEARNING WITH RELATIONAL INDUCTIVE BIASES, Vinicius Zambaldi et.al., ICLR 2019
>
> [5] Learning Reward Machines for Partially Observable Reinforcement Learning, Rodrigo Toro Icarte et.al., NeurIPS 2019
>
> [6] A Reinforcement Learning Algorithm in Partially Observable Environments Using Short-Term Memory, Nobuo Suematsu and Akira Hayashi, NeurIPS 1998
>
> [7] Inverse Reinforcement Learning in Partially Observable Environments, Jaedeug Choi and Kee-Eung Kim, JMLR 2011

---

> > ### Author Response · Authors · 2021-09-03
> > **Re: after rebuttal**
> >
> > We are sorry to see that you lowered your score, but we appreciate your constructive suggestions! We would like to clarify a few points regarding your remaining concerns.
> >
> > > Having read all discussion, I slightly lower my score (6 -> 5). On one hand, the idea is promising, the direction seems important, and the authors addressed many important questions. On the other hand, I agree with kENU that many important details regarding training, ablations, and baselines are needed in the paper. Further, as detailed below, I think the paper needs additional experiments. The proposal about “interweaved hallucinator training with executor learning” should be discussed in the paper.
> >
> > We are glad that the reviewer thinks the idea is promising and the direction is important. We will add the discussions in the rebuttal to our paper to clarify details regarding training, ablations, and baselines. We discuss about the additional experiment suggested by the reviewer below.
> >
> > > However, I have additional concerns about baselines. Currently, the End-to-end baseline, the one with many micro-actor DNNs, is severely handicapped: (1) it does not have access to a hallucinator (mentioned in the response to Wt3P, experiments/craft_nn.yaml#L7, and models/simple_nn.py#L59-66), and (2) it is not recurrent (judging from a quick regex over the supplied codebase), contrary to the provided response to kENU. Point (2) is rather concerning since it is unclear how the non-markovian dependence on the full history is handled.
> >
> > We would like to point out that the options used in our approach are also not recurrent: the actor DNNs in the end-to-end baseline and the actor DNNs in our approach have exactly the same architecture, take the same form of inputs. The only difference is how they are composed to form the full policy. In the end-to-end baseline, each actor DNN is trained to solve a type of task in full. In our approach, each actor DNN is trained to solve one component, which is a part of the whole task. The reason why our approach is able to decompose the full tasks into components is because we do program synthesis -- the synthesized program provides a decomposition of the task. This decomposition is the key reason why our approach (and other program-guided approaches) outperforms non-program-guided baselines; this is also found in the prior work on program-guided RL. Even a recurrent version of the end-to-end baseline, supported with a hallucinator, cannot decompose the tasks and thus the policy into components like the program-guided approaches do. Nevertheless, we agree that including such a baseline could further make our results stronger.
> >
> > As a side note, we did not say that the end-to-end baseline is recurrent anywhere in our response to kENU. Just to clarify here so that it won't be misread by the broader audience.
> >
> > > At the same time, the World Model baseline (recurrent) does not have access to pre-trained options, which makes the comparison unfair. Thus, the baselines do not allow us to separate the effect of good micro-goal options from the synergy between the hallucinator and the synthesizer.
> >
> > We would like to emphasize that the actors in the world models baseline, in our approach, and in the end-to-end baseline, are all trained in the same way (actor-critic with curriculum learning, as discussed in Section 5). The actor DNN in the world models baseline (which is called the Controller Model in the original world models paper) takes the hidden state from the recurrent model (M model) and the map observations as input. There is no difference on how the actors (or options) are trained between our approach and the world models/end-to-end baseline, except the decomposition enabled by program guidance (as discussed above). Therefore, there is no differentiation on ''have access to pre-trained options'' between our approach and the world models baseline.
> >
> > > In light of the claimed benefits of synthesized programs, I think a good baseline/ablation experiment would be to replace the synthesizer with a recurrent neural policy on pre-trained options, which also has access to the hallucinator or an equivalent world model (also pre-trained on a fully observable environment). To fairly judge the benefit of program synthesis here it may be necessary to compare the number of requests each approach makes to the world model.
> >
> > We can think of two interpretations of the approach suggested by the reviewer. The first is to have a recurrent neural policy as the actor (i.e. outputs low-level actions), and this recurrent policy has access to a hallucinator/world model. This baseline still cannot decompose the task/policy into components, therefore it still cannot enjoy the benefit brought by the synthesized programs. Nevertheless, we agree that including this baseline could further make our results stronger.
> >
> > The second approach is to have a recurrent neural policy to choose which component to use, and the actors are the options for each component as in our approach. This recurrent neural policy is essentially a neural program synthesizer -- it predicts what is the next component in the program. Therefore, this is also a program-synthesis-based approach. In fact, we think this is a great direction for future research, exploring neural program synthesizer in the program-guided RL context.
> >
> > > As for the claims of reduced user supervision, I suspect that identifying micro-goals and specifying them as logical predicates still demands that the user thoroughly analyzes the environment (L32). So the toning down of the contributions of the paper seems appropriate. Besides, micro-goals' logical predicates as they are described in the paper do not seem to be very high-level (L35): one has to tinker with the before- and after- states, enumerate positions, and zones in the grid, and compose conjunctive forms (L496). Despite this, I am inclined to agree that such specifications are more basic than a program description of an optimal solution as in Sun et al. (2017).
> >
> > We agree that identifying micro-goals and specifying them as logical predicates still demands that the user thoroughly analyzes the environment. However, this is a one-time effort -- it only needs to be done once for an environment/domain, and can be reused for any new tasks in this domain. In comparison, manually providing guiding programs for each task requires additional user effort for every new task appeared in the future. We believe that reducing a continuous and indefinite sequence of effort to a one-time upfront effort is a significant change.
> >
> > We hope that the above answers can help clarify the remaining concerns by the reviewer. We are happy to answer any additional questions.

---

> > > ### Comment · Reviewer_EBwR · 2021-09-03
> > > **Final comments on experiments**
> > >
> > > To reiterate, I find your idea of _a program synthesizer with a generative model for inputs_ worthy of publication, but still there should be additional experimental verifications. I summarize the experimental questions below.
> > >
> > > I think it is necessary to show in both the World Models and E2E experiments which pieces of the MPPS specifically make it so successful. Is it a world hallucinator (a similarly trained equivalent thereof)? Or is it the access to privilegedly pre-trained micro-goal options? What if such well-trained options make any method better off? Or, finally, is it the _crispness_ of a synthesized program as opposed to a fuzziness of a neural policy?
> > >
> > > For example, the experiment I proposed in the previous reply would seek an answer to the following question: "conditional on _exactly identical_ pre-trained options (fully observed env. with random exploration, etc.), **does the MPPS**, which explicitly synthesizes chains of options using a SATsolver logic, **beat a neural policy**, trained to tug on the provided options based on its implicit recurrent representations".

---

> > > > ### Author Response · Authors · 2021-09-03
> > > > **Re: Final comments on experiments**
> > > >
> > > > Thank you for reiterating your questions. To sum up, as we discussed in our previous response, **the main reason why MPPS (and other program-guided approaches) has a better performance than the world models and the E2E baseline is due to the decomposition of the task/policy**. What the reviewer refers to as ''micro-goal options'' are the results of such decomposition. Being able to decompose a task into these micro-goal options is why our approach is successful. **Importantly, having the guiding programs is what enables such decompositions.** The program specifies the high-level strategy to solve a task, and that enables us to train these micro-goal options at the first place (we need the guiding programs to train these micro-goal options, see L211-223). Therefore, the non-program-guided baselines like the world models and the E2E cannot train such micro-goal options at the first place, and also cannot use them. Furthermore, except the decomposition enabled by programs, the way we train the micro-goal options in our approach and the actor policies in world models and E2E are exactly the same, **so the micro-goal options in our approach is neither more ''privilegedly'' nor more ''pre-trained'' than the world models and E2E**.
> > > >
> > > > Regarding the approach proposed by the reviewer, which uses a recurrent neural policy to select which component to execute next, we think this is a great idea to explore. **This approach is essentially replacing the MaxSAT based synthesizer with a neural program synthesizer**. Being able to train the micro-goal options at the first place means that this approach needs programs to specify the policy structure, therefore **this is also a program-synthesis-based approach**. The recurrent neural policy that tugs on the micro-goal options is solving the problem of neural program synthesis/neural program induction. Neural program synthesis/induction is an active field, we provide some recent work here so that the readers can understand the link:
> > > > - EXECUTION-GUIDED NEURAL PROGRAM SYNTHESIS, Xinyun Chen et.al., ICLR 2019
> > > > - NEURAL PROGRAMMER: INDUCING LATENT PROGRAMS WITH GRADIENT DESCENT, Arvind Neelakantan et.al., ICLR 2016
> > > >
> > > > We think that applying neural program synthesizer approaches in this context of program-guided RL is in itself a very interesting research direction. We thank the reviewer for the suggestion.

---

### Author Response · Authors · 2021-08-12
**Overall Response**

We thank the reviewers for their reviews! We would like to use this place to make an overall response to each of the reviewers.

First of all, we are extremely thankful for reviewer aK3Q and reviewer EBwR to spend so much of their valuable time in providing these positive, thoughtful and insightful reviews! The constructive suggestions on how to improve the paper and the ideas pointing to interesting directions for future research are more than valuable for us. Thank you!

Reviewer Wt3P mainly asked about several clarification questions, in particular the question on why program guidance is beneficial. The main rationale is that programs can specify a high-level structure of the policy, which makes reinforcement learning more efficient for long-horizon problems. Prior work (e.g. Program Guided Agent, Sun et.al., ICLR 2020) has made a solid contribution in demonstrating this. We hope that our detailed response helps clarify your questions.

Reviewer 1PbL mostly asked about what is our targeted problem, what do several concepts mean, what are our experiment settings, etc.. We hope that our answers and pointers to the sections in our paper that describe these points are helpful for the reviewer's understanding.

Finally, we thank reviewer kENU for the detailed review! We hope that our answers help clarify the details on hallucinator training, baselines, and evaluation settings. We also provide answers and additional experimental results to justify design choices we made on training setup, baselines, and assumptions. We hope that our response addresses your questions and concerns.

Please see the individual responses for detailed answers and additional experimental results. We hope that our responses address the questions and concerns from the reviewers, and we are happy to discuss more if the reviewers have further questions!

---

### Decision · Program_Chairs · 2021-09-28

**Decision:**

Accept (Spotlight)

**Comment:**

All reviewers agree that the paper tackles an interesting research problem, ie long-term planning in POMDPs. However, multiple issues were raised and remain after the discussion, among these are clarity of exposition and mostly importantly novelty of contribution. Concerning the latter, introduction of a generative model into the architecture for modeling belief states in partially observed environments has been addressed more thoroughly in the literature before (eg see [1] and references therein) mostly without the need for privileged information during training. The problem of planning in POMDPs has an even richer history, eg the method proposed in [2] would be directly applicable to the setting considered here. However the method is not referenced nor compared to in experimental results. The paper is therefore not accepted in its current form.

[1] Temporal Difference Auto-Encoder. Gregor at al.
[2] Monte-Carlo Planning in Large POMDPs. Silver, Veness.

**Consistency Experiment:**

NeurIPS has a long history of experimentation. In 2014, NeurIPS ran an experiment in which 10% of submissions were reviewed by two independent committees to quantify the randomness in the review process. This year, we repeated a variant of this experiment to see how the quality of the review process has changed over time.  This paper was part of the experiment and was therefore assigned to two committees (consisting of reviewers, an Area Chair, and a Senior Area Chair) that reached independent decisions.  If both committees made the same recommendation, this recommendation was followed. If a single committee recommended acceptance, the paper was accepted (with the exception of a few cases in which the other committee identified what we considered a fatal flaw, e.g., an error in a key result).

This copy’s committee reached the following decision: **Reject**

The other committee assigned to the paper recommended **Accept (Spotlight)**.  You can find the other set of reviews, along with any follow up discussion with the authors here:
https://openreview.net/forum?id=43fmQ-db-yJ